# A Comprehensive Review of Almond Clinical Trials on Weight Measures, Metabolic Health Biomarkers and Outcomes, and the Gut Microbiota

**DOI:** 10.3390/nu13061968

**Published:** 2021-06-08

**Authors:** Mark L. Dreher

**Affiliations:** Nutrition Science Solutions, LLC, 900 S Rainbow Ranch Rd, Wimberley, TX 78676, USA; nss3@sbcglobal.net

**Keywords:** almonds, energy density (ED), nuts, body weight, body mass, central obesity, body mass index (BMI), waist circumference (WC), body fat % (BF%), fat mass (FM), fat-free mass (FFM), low-calorie diets (LCDs), blood lipids, glycemic control, insulin sensitivity, hs-C-reactive protein (hs-CRP), endothelial health, blood pressure (BP), pre-diabetes, type 2 (T2) diabetes, cognitive performance, colonic microbiota, coronary heart disease (CHD), precision nutrition

## Abstract

This comprehensive narrative review of 64 randomized controlled trials (RCTs) and 14 systematic reviews and/or meta-analyses provides an in-depth analysis of the effect of almonds on weight measures, metabolic health biomarkers and outcomes, and the colonic microbiota, with extensive use of figures and tables. Almonds are a higher energy-dense (ED) food that acts like a lower ED food when consumed. Recent systematic reviews and meta-analyses of nut RCTs showed that almonds were the only nut that had a small but significant decrease in both mean body mass and fat mass, compared to control diets. The biological mechanisms for almond weight control include enhanced displacement of other foods, decreased macronutrient bioavailability for a lower net metabolizable energy (ME), upregulation of acute signals for reduced hunger, and elevated satiety and increased resting energy expenditure. The intake of 42.5 g/day of almonds significantly lowered low-density lipoprotein cholesterol (LDL-C), 10-year Framingham estimated coronary heart disease (CHD) risk and associated cardiovascular disease (CVD) medical expenditures. Diastolic blood pressure (BP) was modestly but significantly lowered when almonds were consumed at >42.5 g/day or for >6 weeks. Recent RCTs suggest possible emerging health benefits for almonds such as enhanced cognitive performance, improved heart rate variability under mental stress, and reduced rate of facial skin aging from exposure to ultraviolet (UV) B radiation. Eight RCTs show that almonds can support colonic microbiota health by promoting microflora richness and diversity, increasing the ratio of symbiotic to pathogenic microflora, and concentrations of health-promoting colonic bioactives. Almonds are a premier healthy snack for precision nutrition diet plans.

## 1. Introduction

Compared to other nuts, a serving of almonds provides the highest or among the highest amounts of fiber, protein, monounsaturated and polyunsaturated fats, magnesium, calcium, iron, folate, riboflavin, niacin, vitamin E, phytosterols, flavonoids, and phenolic acids, and are among the lowest in calories, available carbohydrates, and saturated fat [1,2]. An analysis of the United Kingdom’s National Diet and Nutrition Survey 2008–2017 showed that whole almond consumers’ diets had a higher nutrient quality score than non-consumers based on their higher intakes of protein, unsaturated fats, fiber, folate, vitamin E, and magnesium, and lower intakes of total carbohydrates, sugar, and sodium [3]. Additionally, the almond consumers had a lower mean body mass index (BMI) and waist circumference (WC) than non-consumers because when almonds are consumed, they stimulate a series of weight control mechanisms [4,5,6,7,8,9,10]. Meta-analyses of prospective studies have observed that frequent nut intake (≥28 g/day) was associated with lower weight gain and risk of being overweight or obese compared to nut-free diets [11,12,13]. In addition, meta-analyses of prospective cohort studies have observed that nuts support better metabolic health to lower the risk of cardiovascular disease (CVD), total cancer, and lower all-cause mortality and mortality from CVD, stroke, respiratory disease, diabetes, and infections in the general population [14,15,16]. The maintenance of a healthy weight throughout adulthood, especially in early adulthood, limits the number of enlarged fat cells and amount of visceral and ectopic fat which secrete a number of hormonal and inflammatory signals that can damage arteries, heart, liver, muscle, lungs, pancreas, bones, joints, and increase the risk of infectious diseases [17,18,19,20,21,22]. Since there are no specific large prospective studies specifically on almonds, the primary focus of this review is on almond-based randomized controlled trials (RCTs) and human health. Almonds are among the most researched foods, with over two decades of continuous research and more than 175 peer-reviewed published papers on heart and diabetes health, weight management, and other public health concerns. This review of RCTs includes 64 RCTs and 14 systematic reviews and/or meta-analyses on almonds and some other nuts on: weight measures including body mass (weight), fat mass, waist circumference (WC), fat-free mass, body mass index (BMI), percentage body fat (BF%), and visceral fat; metabolic health biomarkers including lipid profiles, blood pressure, insulin sensitivity, vascular reactivity, and oxidative and inflammatory stress; health outcomes including cardiovascular diseases (CVD) and type 2 (T2) diabetes, and colonic microbiota health profiles including α and β diversity, microflora populations, and short-chain fatty acid (SCFA) concentrations. There are also a number of possible emerging almond health outcomes on improved cognitive performance, variable heart rate when under mental stress, and protection against adverse ultraviolet (UV) B radiation on facial aging. The objective of this narrative review is to provide a comprehensive overview of data from all almond RCTs on body weight and composition measures, metabolic health biomarkers and outcomes, and the colonic microbiota, with numerous figures and tables to highlight important RCT design variations and result attributes that are not typically presented with detail in systematic reviews and meta-analyses, which are designed to focus on, for example, the quality of the RCTs analyzed, overall mean, coefficient intervals, dose–response effects, effect sizes, or level of heterogeneity.

## 2. Materials and Methods

This comprehensive narrative review includes 64 RCTs and 14 systemic reviews and/or meta-analyses of RCTs on almonds and some other nuts, which were identified in a literature search of PubMed and Google Scholar from 1 October 2020 to 31 March 2021. All individual RCTs from this search were included in this review. However, for the systematic reviews and meta-analyses used, only more recent papers from 2015 to 2021 were included to assure more current analyses of the almond trials. This review of almond RCTs is divided into three sections. The first section focuses on the effect of almonds on weight measures, including the following search terms: almonds, tree nuts, total nuts, body mass, weight gain, weight loss, adiposity, overweight, obesity, body fat, central obesity, visceral fat, waist circumference (WC), appetite, hunger, satiety, satiation, metabolizable energy (ME), energy density (ED), low-calorie diets (LCDs), and weight control mechanisms. The second section focuses on the effect of almonds on metabolic health biomarkers and outcomes, including the following search terms: chronic diseases, cardiovascular disease (CVD), type 2 (T2) diabetes, metabolic syndrome, blood lipids, glycemic control, insulin sensitivity, oxidative and inflammatory stress, blood pressure (BP), vascular reactivity, cognitive performance, and facial ultraviolet (UV) B radiation protection. The third section focuses on the colonic microbiome with the following search terms: colonic microbiota, almonds, nuts, colonic health, short-chain fatty acids, and diversity.

## 3. Results

### 3.1. Almond RCTs on Weight Measures

#### 3.1.1. Systematic Reviews and Meta-Analyses of RCTs on Almonds and Other Nuts

Six systematic reviews and meta-analyses published since 2018 have evaluated total and/or specific nut-enriched diets compared to control diets for their effects on body mass, fat mass or %, BMI, and WC [23,24,25,26,27,28]. These analyses showed that only almonds had a small but significant mean decrease in body and fat mass compared to control diets to help reduce the risk of being overweight or obese, especially at doses >42.5 g/day and for >6 weeks duration [24,25,26,27,28].

Guarneiri et al. [23], in a 2021 systematic review and meta-analysis of 55 RCTs of all commonly consumed nuts, found that there was no significant overall mean change in body mass, BMI, and WC in diets with 10–100 g nuts/day consumed over 3 to 52 weeks, regardless of the level of dietary substitution guidance provided. However, nut diets with dietary substitution guidance significantly reduced overall mean % body fat (BF%) compared to nut diets without dietary substitution guidance (Table 1). These results indicated that the consumption of nuts, even in relatively large amounts as snacks or with meals, does not significantly increase body mass, BMI, or WC, but the dietary substitution of nuts for lower-quality snacks or foods can significantly lower BF%. In this analysis, almonds accounted for 36% of the overall effect size in studies without substitution guidance and 21% of the overall effect size in studies with substitution guidance 

Li et al. [24], in a 2018 meta-analysis (62 RCTs, 7184 individuals), evaluated the mean differences between specific nut and control diets on body mass (weight), BMI, and WC. Almonds were the only nut that significantly lowered mean body mass by −0.56 kg based on 20 RCTs compared to an overall mean body mass loss of −0.22 kg (Figure 1) for all nuts studied, where almonds accounted for 48% of the overall nut body mass loss effect. In addition, almond diets lowered mean BMI by −0.49 kg/m^2^ and waist circumference (WC) by −2.4 cm compared to a lowered overall mean nut BMI by −0.16 kg/m^2^ and WC by −0.51 cm, where almonds accounted for 34% and 29% of this overall mean nut effect size. There was a relatively high heterogeneity between the different types of nuts on body mass, BMI, and WC. The investigators believed that this heterogeneity and the low number of RCTs for some nuts made the pooled estimates too limited to conduct a complete stratified analysis by nut subtype, and the meta-analysis had geographical limitations as most of the RCTs were from North America or Europe. 

In 2020, almonds, walnuts, and pistachios each had an independent meta-analysis published with important updated insights on the effects of these nuts on changes in body mass, fat mass or %, BMI, and WC compared to control diets [25,26,27]. A comparison of the findings of body and fat mass between these nuts are summarized in Figure 2A and comparison findings on BMI and WC between these nuts are summarized in Figure 2B. Only the intake of almonds significantly decreased both body and fat mass, and the effects of these nuts on BMI and WC were variable.

Eslampour et al. [25], in a systematic review and meta-analysis of 28 almond RCTs, evaluated the mean differences between almond and control diets on weight measures. Almond intake significantly reduced body mass by −0.38 kg (−0.65, −0.10 kg, *p* < 0.007) and fat mass by −0.58 kg (−0.87, −0.28 kg, *p* < 0.001), and tended to decrease BMI by −0.30 kg/m^2^ (−0.67, 0.06 kg/m^2^, *p* = 0.101) and WC by −0.60 cm (−1.28, 0.06 cm, *p* = 0.078) and increase fat-free mass by 0.23 kg (−0.04, 0.50 kg, *p* = 0.097). Subgroup analyses showed that the level of almond weight loss was significantly better after durations of ≥6 weeks and doses of ≥45 g/day. The investigators concluded that “almonds might have a considerable favorable effect on body and fat mass, and their consumption could be encouraged as part of a healthy diet in order to reduce the risk of overweight and obesity and sustain normal weight”.

Fang et al. [26], in a systematic review and dose–response meta-analysis of 27 walnut RCTs, assessed the mean differences between walnut and control diets on weight measures. Walnut intake insignificantly altered body mass by 0.08 kg (−0.03, 0.198 kg, *p* = 0.16), fat mass by 0.28% (−0.49, 1.06%, *p* = 0.48), BMI by −0.4 kg/m^2^ (−0.24, 0.16 kg/m^2^, *p* = 0.70), and WC by −0.19 cm (−1.03, 0.64 cm, *p* = 0.65). However, dose–response analysis suggested a non-linear response with a limited subset of trials (*n* = 6) and supported that walnut intake of <35 g significantly lowered body mass, BMI, and WC, but not body fat. The investigators concluded that “to date, there is no discernible evidence to support walnut intake for improving anthropometric indicators of weight loss”.

Xia et al. [27] examined, in a systematic review and meta-analysis of 11 pistachio RCTs, the mean differences between pistachio and control diets on weight measures. Pistachio intake significantly lowered BMI by −0.18 kg/m^2^ (−0.26, −0.11 kg/m^2^, *p* < 0.001) and tended to decrease body mass by −0.22 kg (−0.50, 0.07 kg, *p* = 0.14) and increase WC by 0.76 cm (−0.11, 1.63 cm, *p* = 0.087). The researchers concluded that “a diet with pistachios reduced BMI and had no significant effects on body weight or WC”.

Lee-Bravatti et al. [28], in a 2019 meta-analysis of RCTs, evaluated the difference between almond and control diets on body mass and BMI. Almond-enriched diets with >42.5 g almonds/day and trial durations > 6 weeks significantly lowered mean body mass, but BMI was not significantly altered by almond dose or trial duration (Table 2).

#### 3.1.2. Individual Almond RCTs 

Habitual Diets with Added Almonds

Thirteen RCTs evaluated the effects of habitual diet with added almonds compared to almond-free control diets on changes in body mass and composition, meal and total daily energy intake, energy density (ED) measurement methods, displacement of other snacks and discretionary foods, and hunger and appetite control [5,6,7,29,30,31,32,33,34,35,36,37,38,39]. Although nuts are higher in ED (e.g., rich in lipids) with the potential to increase the risk of unintended body mass gain, RCTs consistently demonstrate that increased adiposity is not a risk with daily almond consumption. 

Fraser et al. [29] conducted a crossover RCT which randomly assigned 81 healthy adults (43% women, mean age 50 years, mean BMI 26) to add an average of 54 g raw or roasted almonds/day (about 320 kcals/day, 40–50 nuts) to their habitual diet or to continue their habitual diet without added almonds as a control for six months. Compared to the control diet, the almond diet did not statistically or clinically alter body mass, with an overall gain of + 0.4 kg compared to the predicted body fat mass gain of 6.4 kg based on the added almond calories (almonds provided an average of 57,500 kcals for subjects during this study). A sub-group analysis showed that almond effects on body mass varied by sex and BMI: women were better able to maintain baseline body mass than men, and baseline lower or medium tertile BMI subjects gained body mass and higher BMI individuals (>28 BMI) actually lowered their body mass to varying levels. However, according to the investigators, all the body mass changes were relatively small and probably not biologically significant. Dietary analyses from food recalls showed that 54% of the extra almond energy was displaced by reduced intake of other foods, and food diaries showed 78% displacement. Jaceldo-Siegl et al. [30] showed that the addition of almonds to the habitual diet made for a healthier and more weight-controlled diet by significantly increasing levels of unsaturated fat, protein, fiber, magnesium, and alpha-tocopherol, and decreasing level of trans-fat, animal protein, cholesterol, sodium, and sugar.

Hollis and Mattes [31], in a crossover RCT, randomized 20 healthy women (mean age 24 years, mean BMI 26) into their habitual diet plus 60 g raw almonds/day (2.1 servings, 344 kcal) or an almond-free habitual diet for 10 weeks without any dietary advice for weight maintenance. Despite a predicted body fat gain of 3 kg, the almond-enriched diet did not significantly result in change in body mass (kg), body fat (%), fat mass (kg), fat-free mass (kg), or total food intake compared to the almond-free control diet (*p* > 0.05 for all, Figure 3). The biological processes by which added almond intake dissipated metabolizable energy (ME) included a 74% dietary energy compensation by reduced food intake from other sources, 7% increased fecal excretion of macronutrients, and 24% increased resting energy expenditure.

Lovejoy et al. [32] conducted a clinical trial in which 20 healthy individuals (10 men and 10 women, mean age 25 years, mean BMI 23) consumed 100 g almonds/day (3.6 servings) (579 kcals) as whole almonds alone or in a variety of foods, such as trail mix, cookies, or muffins for 4 weeks. Compared to baseline, body mass increased slightly but significantly after 4 weeks of almond intake by 0.3 kg for women and by 0.9 kg for men (*p* = 0.006), which was much lower than the predicted increase of about 2.1 kg of body fat based on the amount of added almond consumed.

Tan and Mattes [33], in a parallel RCT, randomized 137 overweight or obese adults at risk for type 2 diabetes (65% women, mean BMI > 27, mean age 29 years) into 5 diets: nut-free usual diet (control), morning snack, afternoon snack (43 g almonds each), breakfast with almonds, and lunch with almonds (43 g almonds each) for 4 weeks without specific dietary guidance. There were no significant differences in body mass, body fat mass, BMI, and WC, between the almond-supplemented groups and the control group. Despite the additional 250 kcal/day from almonds, the daily energy intakes of the almond diet groups were not significantly different from the control diet group or baseline. Eating almonds either with meals or as snacks resulted in lower hunger levels at the next meal. Additionally, the consumption of almonds with breakfast was better at reducing post-meal hunger ratings than when almonds were consumed with lunch, but afternoon almond snacking was the most effective at suppressing hunger. 

Sweazea et al. [34], in a parallel RCT, randomly assigned 21 adults with type 2 diabetes (57% women, mean age 56 years, mean BMI 35) to a habitual diet with ≤2 servings of non-almond nuts (control) or a habitual diet plus 43 g (250 kcals) of almonds 5–7 times weekly for 12 weeks. Compared to the control diet, the almond diet insignificantly increased overall mean body mass by 1.2 kg (*p* = 0.973), BMI by 0.4 kg/m^2^ (*p* = 0.973), WC by 1.3 cm (*p* = 0.809), and body fat by 2.8% (*p* = 0.468), which was less than the estimated increase in body mass of 6 kg predicted for the level of almond energy added to the diet over 12 weeks. 

de Souza et al. [35], in a parallel RCT, randomized 60 obese women (age 20–59 years, mean BMI 33) to habitual, isocaloric diets enriched with 20 g/day of roasted baru almonds (109 kcals) or a baru almond-free control diet for 8 weeks. Compared to control diets, baru almond-enriched diets significantly reduced WC by −2.45 cm (−3.90, −0.23 cm, *p* = 0.03), but there were no significant differences between the diets for body mass, BMI, fat mass, and lean mass (*p* ≤ 0.95), as shown in Figure 4. The baru almond diet reduced abdominal adiposity in overweight/obese women compared to isocaloric control diets. 

Bento et al. [36], in a crossover RCT, randomized 20 mildly hypercholesterolemic subjects (8 men and 12 women, mean age 35 years, mean BMI 23) into their habitual diet plus 20 g (109 kcals) of baru almonds or a corn starch placebo capsule for 6 weeks. The baru almond-enriched diet tended to reduce body mass, BMI, and body fat (%) compared to the placebo (Figure 5).

Jamshed et al. [37], in a parallel RCT, randomized 150 adults with CHD (mean age 60 years, 25% women, mean body mass 76 kg) into a habitual diet plus 10 g of Pakistani or American almonds/day (soaked in water overnight, skin removed, and consumed before breakfast) or a control habitual diet (no added almonds or other nuts) for 12 weeks. Mean body mass was significantly reduced by −0.6 kg in almond groups compared to the control diet (*p* < 0.05).

Spiller et al. [38], in a parallel RCT, randomized 45 hyperlipidemic adults (73% women, mean age 53 years, mean body mass 66 kg) to the usual diet plus 630 kcals/day from: (1) raw whole and ground almonds (100 g/day), (2) olive oil (48 g), cottage cheese (113 g), and rye crackers (21 g), or (3) cheddar cheese (85 g), butter (28 g), and rye crackers (21 g) to match the total fat level for each diet. There were no significant differences in body mass for these diets after 4 weeks (*p* > 0.05). 

Cohen and Johnston [39], in a parallel RCT, randomized 13 obese subjects with T2 diabetes (54% men, mean age 66 years, mean BMI 37) to their usual diets plus a serving of almonds (28 g/day, 164 kcals, 24 almonds) or a serving of cheese sticks (160 kcals) for 5 days per week for 12 weeks. Snacking on almonds significantly lowered BMI and tended to lower body mass compared to string cheese (Figure 6).

Novotny et al. [5] conducted a dose–response, crossover, controlled feeding RCT including 18 healthy adults (10 men and 8 women, mean age 56 years, mean BMI 27) who were randomized into a 3-arm typical American diet with 0.0, 42, and 84 g of whole raw almonds/day for 18 days to measure the energy density (ED) of whole almonds in the human diet. This study found that the mean ED for whole raw almonds was 4.6 ± 0.8 kcals/g or 129 kcals/serving compared to the USDA estimated Atwater ED 6.0 kcals/g or approximately 164 kcals/serving. The Atwater ED estimates when applied to whole raw almonds were overestimated by 23%. This lower ED value was primarily associated with more reduced almond fat bioavailability than that projected by the Atwater estimates. The fat digestibility was found to be decreased by 5% for 42.5 g raw almonds and by 10% for 84 g almonds (*p* < 0.0001).

Gebauer et al. [6] conducted a 5-period crossover, controlled feeding RCT in 18 healthy adults (56% men, mean age 57 years, mean body mass 89 kg) who were randomized into a typical American diet plus 42.5 g whole raw almonds, whole roasted almonds, chopped roasted almonds, or almond butter (data not shown), plus a control diet for 3 weeks to directly measure ED in kcals/g. The clinically measured EDs for whole raw almonds (4.42 kcals/g), whole roasted almonds (4.86 kcals/g), and chopped roasted almonds (5.04 kcals/g) were significantly lower than those predicted with Atwater factors, by 25%, 19%, and 17%, respectively (*p* < 0.001; Figure 7). The clinical ED of whole raw almonds was lower than that of roasted whole and roasted chopped almonds (*p* < 0.05), which was due to lower hardness of whole roasted and chopped almonds compared to whole raw almonds (*p* < 0.05). The harder whole raw almonds fractured into fewer, larger particles, inhibiting the release of lipids for gut bioavailability. Almond butter had an ED of 6.53 kcals/g which was basically the same as the Atwater estimated ED value of 6.62 kcals/g, because almond butter fat is readily bioavailable, and a better fit for the Atwater estimate.

Hull et al. [7], in a crossover RCT, randomly assigned 32 healthy women (mean age 48 years, mean BMI 23) to consume a standard breakfast followed by a mid-morning snack of a 100 mL water control and no almonds, 28, or 42 g of almonds with 100 mL of water and subsequent ad libitum meals at lunch and dinner to test for subjective appetite ratings of hunger and fullness, and both meal and total energy intake. The intake of 28 g raw almonds as a mid-morning snack significantly reduced lunch energy intake (*p* = 0.016) but not dinner energy intake (*p* = 0.28), and the 42 g mid-morning almond snack significantly lowered both lunch and dinner energy intake (*p* ≤ 0.001) (Figure 8). The almond snack decreased energy intake in the subsequent lunch and dinner meals in a dose–response manner for a net insignificant change of the daily total energy intake compared to the 0 g almond control. There was an almond dose-dependent increased subjective acute satiety rating for lower appetite in subsequent meals.

Habitual Diets with Almond and Carbohydrate-Rich Snacks.

Ten RCTs evaluated the effects of “snacking” on 50–84 g almonds/day compared to isocaloric carbohydrate-based snacks consisting of cereal bars, cookies, crackers, mini-muffins, biscuits, potato chips, or a variety of snack choices, including low-fat potato chips, bagels, rice cakes, pretzels, vanilla cookies, dried mango slices, or pudding cups, with the usual diet in individuals ranging from 18 to >60 years for 4–20 weeks to examine changes in hunger, satiety, energy intake, and body weight and composition [40,41,42,43,44,45,46,47,48,49]. Snacking on almonds was more effective than snacking on savory crackers for acutely reducing the overall hunger drive, but there was no overall 24 h difference in energy intake [40]. An almond pre-meal 18.5 g dose before each meal significantly reduced body fat mass and visceral fat compared to cookies between baseline and 8 weeks [42]. In RCTs of 4–12 weeks duration, there were generally no or limited differences in body mass or composition measures between almond and carbohydrate-rich snacks [41,44,45,46,47,48,49]. However, in RCTs of ≥16 weeks, almond snacking tended to improve both body fat and lean mass compared to high-carbohydrate control snacks [42,43]. 

Hollingworth et al. [40], in a crossover RCT, randomized 42 women (mean age 26 years, mean BMI 22) who consumed a standard breakfast into a water control or one of two mid-morning matched energy density snacks: raw almonds (53 g) or a savory cracker. They were evaluated for hunger ratings over 10 h. Almonds as a mid-morning snack significantly reduced hunger compared to the savory cracker and water control (Figure 9). Additionally, almond snacking compared to the crackers reduced the desire to eat, reduced hedonic wanting for high-fat foods, and reduced prospective energy consumption. The satiating efficiency immediately after intake was greater for almonds compared to crackers (*p* < 0.05). The satiety efficiency decreased over 120 min. This study showed that the addition of almonds to the diet as a mid-morning snack reduced overall hunger and reduced desire for high-fat foods compared to an energy-matched savory cracker snack. 

Zaveri et al. [41], in a parallel RCT, randomized 45 healthy men (mean age 40 years, mean BMI 30) to their habitual diet with the following added snacks: almonds (56 g, 3.9 g available carbs, 7.0 g fiber, 31 g fat, 12 g protein, 343 kcals) or cereal bar (60 g weight, 44 g available carbohydrate, 4.7 g fat, 3.0 g protein, 227 kcals) for 12 weeks. There were no significant differences in body mass, BMI, WC, or BF% between the almond and cereal bar groups but there were changes in waist–hip ratio and eating frequency. The cereal bars significantly increased the waist–hip ratio by 11% after 6 weeks and 22% after 12 weeks (*p* ≤ 0.01), which could be an early indicator of metabolic syndrome risk. The almond snack significantly increased relative eating frequency from baseline by 23.5% compared to cereal bars at 12 weeks (*p* ≤ 0.01) without resulting in higher energy intake or weight gain, which is an indicator of the almond effect on energy compensation via food displacement.

Liu et al. [42], in a parallel RCT, randomized 169 healthy individuals (54% women, mean age 26 years, mean BMI 23) to the usual diet with: (1) 18.5 g almonds before each meal (pre-meal), (2) 56 g almonds as a snack consumed between meals, or (3) an isocaloric cookie (control) consumed between meals for 8–16 weeks. There were no significant differences in body mass or BMI from baseline among the three groups. The effects of almond and cookie intake on body fat measures for both 8 and 16 weeks are summarized in Figure 10A,B. Pre-meal intake of almonds significantly reduced body fat mass (*p* = 0.025) and visceral fat mass (*p* = 0.000) at both 8 and 16 weeks and BF (%) after 16 weeks (*p* = 0.032) compared to the cookie control. Snacking on almonds significantly reduced body fat (%) compared to the cookie control after 16 weeks (*p* = 0.032) without significantly changing total body mass or BMI. 

Liu et al. [43] extended the previous trial [42] to 20 weeks with 57 individuals from the almond (56 g) between meal snack group and 28 individuals from the isocaloric cookie control group (52% men, mean age 27 years, BMI 23). Changes in body mass and composition for almonds and cookies from baseline to week 20 are summarized in Figure 11. Snacking on almonds for 20 weeks improved body composition in healthy young adults compared to the cookie control. The cookie control significantly increased body weight and BMI (*p* < 0.001), body fat mass (*p* = 0.027), and waist–hip ratio (*p* = 0.035) from baseline to 20 weeks. Compared to the cookie control, almond snacking tended to decrease body mass (*p* = 0.058), visceral fat level (*p* = 0.056), and WC (*p* = 0.064). Total body protein, fat-free mass, soft lean mass, and skeletal muscle mass were significantly increased after 20 weeks of almond intake (*p* < 0.001 for all). Over 8 to 20 weeks, almond snacking tended to maintain body mass, but cookie snacking significantly increased body mass (Figure 12) and almond snacking significantly lowered WC, while cookies significantly increased WC (Figure 13). 

Dhillon et al. [44], in a parallel RCT, randomized 73 healthy college freshman students (mean age 18 years, 58% women, mean BMI 26) to their usual diet plus snacking on roasted almonds (57 g/day; 364 kcal) or Graham crackers (78 g/day; 338 kcal/day). Anthropometric and appetite changes were determined from baseline to 4 and 8 weeks for almond and Graham cracker snacks. Both snacks significantly increased body mass and fat-free mass from baseline to 8 weeks (*p* < 0.05). Total body mass, trunk mass, trunk fat-free mass, and WC were not significantly different between baseline and week 8. For both snacks, mean 24 h hunger, desire to eat, and prospective consumption ratings significantly decreased from baseline to 4 weeks, with no difference after 8 weeks except for an increase in prospective consumption ratings. The 24 h fullness ratings did not significantly differ over the 8-week trial. 

Dikariyanto et al. [45], in a parallel RCT, randomly allocated 107 subjects with above-average risk of CVD (56 years, 70% women, mean BMI 27) to consume the typical United Kingdom diet with one of two snacks: (1) dry roasted almonds (63 g, 1/4 cup) or (2) sweet and savory mini-muffins as the control for 6 weeks, with dietary advice from a dietitian and instructions to only consume study snacks and to avoid the consumption of additional nuts or nut products during the study and maintain baseline fruit intake. There were no significant differences between the muffin control and almond snacks in BF (%), BMI, WC (cm), visceral fat volume (%), or liver fat (%). The almond group improved diet nutrition quality compared to the muffin group by reducing energy from starch by −7.0% and free sugar by −3.0%, lowering sodium by 671 mg/day, and increasing unsaturated fat: saturated fat ratio by 1.3, magnesium by 150 mg/day, vitamin E by 15 mg/day, and fiber by 7.4 g/day.

Jung et al. [46], in a crossover RCT, randomized 84 “healthy” overweight/obese individuals (87% women, mean age 52 years, mean BMI 25) into a typical high-carbohydrate Korean diet (64% energy from carbohydrates, 22% energy from fat, and 15% energy from protein) plus daily snacking on roasted almonds (56 g, 340 kcals) or an isocaloric cookie (70 g, 340 kcals) for 4 weeks. Neither snacking on almonds nor cookies altered body or body fat mass, waist circumference (WC), or BMI, compared to baseline. However, the overall nutritional status was improved with almond snacking by decreasing % energy from carbohydrate by 14%, and increasing % energy from monounsaturated fat by 192%, polyunsaturated fat by 85%, vitamin E by 103%, and fiber by 12%. Bowen et al. [47], in a parallel RCT, randomized 76 adults with elevated risk for or with type 2 (T2) diabetes (61 years, 59% male, mean BMI 34) to their usual diet: plus 2 servings of raw almonds (56 g/day) or 2 servings of Arnott’s biscuit snack (energy-matched) for 8 weeks, with individual advice on substituting their usual between meal snacks with their intervention snacks to daily minimize the caloric changes. There were no significant effects on body mass, fat mass, WC (cm), fat-free mass, muscle mass, visceral fat, or liver fat for either snack (*p* > 0.7 for all) (Table 3). The almond diet had 8.7% less energy from carbohydrates, 4.1% more energy from fat, 2.7% more energy from protein, and 8 g/day more fiber than the biscuit diet.

Palacios et al. [48], in a crossover RCT, randomized 54 adults with prediabetes (48% women, mean age 48 years, mean BMI 31, 61% completers) into their usual diet plus snacks consisting of 84 g raw almonds/day (480 kcals) divided into two servings, or an isocaloric variety of carbohydrate snacks including baked chips, pretzels, dried mangos, pudding, cookies, mini-bagel, or rice cakes for 6 weeks. The subjects were instructed to use food substitution in their usual diet during each diet arm to maintain baseline energy intake. There were no significant changes from baseline in energy intake, body mass, or WC (cm) between the diets over 6 weeks (*p* > 0.05). The almond diet significantly decreased % energy from sugar and total carbohydrates, and significantly increased fiber and % energy from unsaturated fat and protein compared to the control carbohydrate-rich snack diet. 

Coates et al. [49], in a parallel RCT, randomized 151 individuals (mean age 65 years, mean BMI 30) into 15% of energy snacks either from raw almonds (estimated 52 g based on a 2000 kcal diet) or isocaloric carbohydrate-rich biscuit or potato chip snacks incorporated into their habitual diets for 12 weeks. Subjects were advised to substitute almond or control snacks for discretionary foods and not to add foods to their usual diet. All subjects were evaluated every three weeks and weighed to check for weight stability. No differences in body mass, BMI, waist (cm), or body fat (%) between the two diets were observed. There was no significant difference in energy intake between the diets, but the almond diet had 20% more energy from fat, primarily from healthy unsaturated fat and a small reduction in saturated fat, lower energy from total carbohydrates and higher fiber intake, plus higher levels of vitamin E, potassium, magnesium, calcium, iron, and riboflavin. 

Healthy Diets with Almonds

Eleven clinical trials evaluating the effects of incorporating almonds (30–100 g/day) into healthy recommended diets for 3–12 weeks on body mass or composition found that almond-enriched diets generally supported improved body mass and composition compared to control diets [50,51,52,53,54,55,56,57,58,59,60,61].

Sabate et al. [50], in a dose–response, crossover RCT, randomly assigned 25 healthy subjects (56% men, mean age 41 years, overweight) to isocaloric National Cholesterol Education Program (NCEP) Step 1 diets containing 0%, 10%, and 20% of total energy from almonds for 4 weeks each, after a 2-week run-in Western diet. There were no significant body weight changes between the diets with the inclusion of 0%, 10%, or 20% energy from almonds (*p*-trend = 0.11).

Jenkins et al. [51,52], in a dose–response, crossover RCT, randomized 27 overweight subjects (56% men, mean age 64 years, mean BMI 26) into NCEP Step 2 diets supplemented with 1 of 3 daily 423 kcal snacks: whole raw almonds (73 g), muffins control (147 g), or a half portion of each. Subjects were counseled on strategies to facilitate weight maintenance. The mean body weight differed by ≤300 g between all the diets from baseline to 4 weeks. The half-dose muffin and almond snack subjects lost significantly more weight than the full muffin snack (*p* < 0.01). 

Berryman et al. [53], in a 2-period, crossover, controlled-feeding RCT, had 48 healthy adults with mildly elevated cholesterol (54% women, mean age 50 years, mean BMI 26) consume 2100 kcal cholesterol-lowering nut-free diets with added daily snacks of raw almonds (42.5 g almonds, 250 kcals) or an isocaloric banana muffin (106 g plus 2.7 g butter, 270 kcals) for 6 weeks. Compared to the muffin control snack, the almond snack significantly reduced from baseline: WC (cm) (−0.8 cm ± 0.3), abdominal mass (−0.19 kg ± 0.08), abdominal fat mass (−0.07 kg ± 0.03), and leg fat mass (−0.12 kg ± 0.05) (*p* = 0.02 for all), despite no significant body mass differences between the two diets. The % change in body weight measures from baseline to 6 weeks are summarized in Figure 14. 

Li et al. [54], in a crossover RCT, randomized 20 adults with T2 diabetes (45% male, mean age 58 years, mean BMI 26) into two isocaloric NCEP Step 2 diets: (1) a nut-free control or (2) roasted almonds (56 g/day average), replacing 20% of the energy of the control diet for 4 weeks. Compared to the control diet group, the almond diet significantly reduced mean body fat mass by −0.8% (*p* = 0.002) without altering total body mass or BMI. Compared to the control, the almond diet had 9.8% less energy from carbohydrate (*p* ≤ 0.001), 9.9% more energy from fat (*p* ≤ 0.001, mainly unsaturated fat), and increased fiber intake of 2.3 g/day (*p* < 0.05).

Chen et al. [55], in a crossover RCT with controlled feeding, randomized 33 patients with T2 diabetes (61% women, mean age 55 years, mean BMI 25) into two isocaloric NCEP Step 2 diets: (1) nut-free control diet or (2) roasted almond diet (60 g/day average), replacing 20% of the energy from the control diet for 12 weeks. Body mass was monitored weekly in order to adjust energy intake to maintain body mass during the study. Compared to the control diet, the almond diet insignificantly reduced mean body mass (−0.7 kg, *p* = 0.241), BMI (−0.4, *p* = 0.148), WC (−0.6 cm, *p* = 0.477), and waist–hip ratio (−0.01, *p* = 0.769). There was no significant difference in total energy intake between the diets. The almond group reduced the % energy from carbohydrates by 8.7% (*p* < 0.001) and increased % energy from fat by 9.2%, primarily from monounsaturated fat (*p* < 0.001), and increased fiber by 6.1 g/day (*p* = 0.004).

Gulati et al. [56], in a pre–post-clinical trial, assigned 63 adults with T2 diabetes (54% men, mean age 46 years, mean BMI 29) to a 3-week run-in diet based on the dietary guidelines for Asian Indians followed by an isocaloric modified run-in diet enriched with 20% energy from raw almonds (about 56 g/day average) for 24 weeks. There was a trend for lower body mass and BMI (*p* < 0.192), and significant reductions in WC and waist–height ratio (*p* ≤ 0.05) for the almond intervention in compliant subjects, which were 80% of all enrolled subjects. 

Spiller et al. [57], in a parallel RCT, randomized 38 free-living, hypercholesterolemic adults (70% women, mean age 61 years, mean weight 68 kg) into a heart-healthy diet including 100 g/day of raw almonds, roasted almonds, or almond butter for 4 weeks. The subjects were advised by nutritionists on how to substitute almonds for other foods and to keep total caloric intake similar to the baseline diet. The consumption of 100 g of almond products did not alter body weight (kg) from baseline with any of these almond sources.

Ruisinger et al. [58], in a parallel RCT, randomized 48 hypercholesterolemic individuals receiving statins (50% men, mean age 61 years, mean BMI 29) to an NCEP Step 3 diet with 100 g raw almonds/day or a control Step 3 diet with dietary counseling for 4 weeks. The almond diet insignificantly increased body weight by 1.0 kg (*p* = 0.145) and BMI by 0.3 kg/m^2^ (*p* = 0.262) compared to the control diet. 

Ren et al. [59], in a parallel RCT, randomly assigned 45 subjects with T2 diabetes (44% men, mean age 72 years, mean BMI 23) to one of two isocaloric diets: (1) an almond-based low-carbohydrate diet (56 g almonds replacing 150 g carbohydrate foods) or (2) a control low-fat (25% energy from fat) diet for 12 weeks with dietary counseling. The almond-based low-carbohydrate diet significantly reduced body weight by −7.3 kg and BMI by −0.5 (*p* < 0.042) compared to the low-fat diet, which insignificantly lowered body weight by −0.5 kg and BMI by −0.16 (*p* < 0.211). 

Damasceno et al. [60] conducted a crossover RCT with 18 hypercholesterolemic adults (mean age 56 years, mean BMI 26) after giving them general guidance for a Mediterranean diet. The study began with a run-in diet for 4 weeks, after which the subjects were randomized into 1 of 3 diets with healthy fat sources by replacing 40% of the run-in diet fat with: (1) virgin olive oil (35–50 g/day), (2) walnuts (40–65 g/day), or (3) raw almonds (50–75 g/day) for 4 weeks each for all subjects. There were no significant differences between the groups (*p* < 0.601) in body weight as all three of these diets lowered mean body weight by −1.8 to −2.0 kg compared to the run-in diet.

Richman et al. [61], in a crossover RCT, randomized 22 postmenopausal women with T2 diabetes (mean age 62 years, mean BMI 29) into a recommended dietary pattern for T2 diabetes with 30 g/day of almonds or sunflower seeds for 3 weeks. Compared to baseline, both almonds and sunflower seeds significantly reduced body mass and BMI (*p* < 0.01), with no significant difference between them (*p* > 0.540).

Low-Calorie Diets (LCDs) with Almonds

Six RCTs evaluated the effects of almond-based LCDs compared to nut-free control LCDs. Those trials with energy deficits of 250 to 500 kcals/day found very generally small significant or no significant differences in body mass and composition between the control and the almond LCDs [62,63,64]. RCTs in the range of 1000 kcal deficits showed that almond LCDs significantly reduced body mass, BMI, body fat mass, waist, hip, and/or waist–hip ratio compared to control LCDs [65,66,67].

Wien et al. [62], in a parallel RCT, randomly assigned 65 adults with prediabetes (74% women, mean age 54 years, mean BMI 30, mean waist 96 cm) to the American Diabetes Association (ADA) diet containing 20% of energy from roasted almonds (60 g/day) or a control ADA diet without almonds for 16 weeks. This included individualized dietitian counseling to help subjects exchange added almonds for other foods or snacks with a prescribed target energy of 250–500 kcals/day deficit in subjects with BMI > 25 kg/m^2^ (78% subjects). There were no significant differences in body mass, BMI, or WC between the groups during this ADA trial over 16 weeks (*p* < 0.6). The authors’ post-trial analysis for the less than expected weight loss noted that it was likely due to an over-estimation of the subjects’ usual physical activity levels in the initial counseling session. 

Foster et al. [63], in a parallel RCT with 123 overweight and obese subjects (90% females, mean age 47 years, BMI 34), randomly assigned the subjects to 1 of 2 LCDs: (1) enriched with whole raw and roasted almonds (56 g/day consumed in two servings) or (2) a nut-free LCD. The targeted energy deficit was approximately 500 kcal/day (1200–1500 kcal/day for women and 1500–1800 kcal/day for men) with standard behavioral methods of hypocaloric weight loss for 18 months. The weight measures for the two LCDs are summarized in Table 4. Those in the almond group lost slightly but significantly less body mass than did those in the nut-free group at 6 months, but there were no differences at 18 months. The loss in fat mass for the nut-free group was nearly statistically significant at 6 months, but no significant differences in changes in fat mass were found between the groups at 18 months. Although there were no significant differences in lean mass between the almond and control groups, the almond group tended to lose less lean mass by 1 kg at 18 months compared to the nut-free control group (*p* = 0.09). 

Dhillon, Tan, and Mattes [64], in a parallel RCT, randomized 86 healthy adults (70% women, mean age 34 years, mean BMI 30) to 1 of 2 LCDs (targeted 500 kcal deficit/day) diets: (1) roasted almond-enriched diet (15% energy or estimated 38 g almonds/day) or a nut-free control diet for 12 weeks. Although the weight, visceral fat, WC (cm), and sagittal abdominal fat reductions did not significantly differ between the almond-enriched and control groups within either the intention-to-treat (ITT) or complier (58% of subjects) analyses, the almond group tended to lose more body mass and visceral adipose tissue (*p* = 0.10) than the nut-free group in the complier analysis. Additionally, the complier analysis showed a significant decrease in percentage of total and trunk fat mass, and a significant increase in percentage of truncal and total fat-free mass in the almond-enriched diet vs. the control diet (*p* < 0.05). The complier analysis more closely reflected better almond diet adherence and related efficacy than the ITT, which had a compliance rate of 65% for the almond and 67% for the control diet. 

Wien et al. [65], in a parallel weight loss RCT, randomly assigned 65 obese and metabolic syndrome middle-aged adults (57% women, mean age 55 years, mean BMI 38) into 1 of 2 diets: (1) a formula-based LCD of 1000 kcals/day, including 84 g almonds (485 kcals/day with 32% energy from carbohydrates, 29% energy from protein, 39% energy from fat, and 20 g fiber), or (2) self-selected complex carbohydrate diets, with 53% energy from carbs, 29% energy from protein, 18% energy from fat, and 32 g fiber, for 6 months under free-living conditions for 24 weeks. The almond LCD had greater % reduction in body mass and BMI (*p* < 0.001), and WC and fat mass (*p* < 0.05) than the carbohydrate LCD (Figure 15). Ketone levels increased significantly in the almond LCD by 260% compared to no increase in the carbohydrate LCD (*p* < 0.02). The carbohydrate group experienced a weight loss plateau at week 16, which did not occur with the almond LCD group. 

Abazarfard et al. [66], in a parallel weight loss (1000 kcals deficit) RCT, randomized 108 overweight and obese Iranian women (mean BMI 30, mean age 43 years, 93% completers) into 1 of 2 LCDs, with: (1) 50 g raw almonds/day divided into two servings or (2) a nut-free control for 12 weeks. Compared to the nut-free LCD, the almond-enriched LCD significantly lowered body mass by −2.4 kg, BMI by −1.0 kg/m^2^, WC by −8.7 cm, and waist–hip ratio by −0.1 (*p* < 0.001). Additionally, Abazarfard et al. [67], in a parallel weight loss (1000 kcals deficit) RCT, showed similar results to their previous trial [66]. They also found that the almond LCD reduced liver enzyme concentrations in obese women (*p* < 0.001), which is likely associated with less accumulation of liver fat. 

### 3.2. Almond RCTs on Metabolic Health Biomarkers and Outcomes

#### 3.2.1. Systematic Reviews and Meta-Analyses of Almonds and Some Other Nut RCTs

Del Gobbo et al. [68], in a 2015 systematic review and dose–response meta-analysis, assessed 61 tree nut clinical trials in 2582 individuals for the overall mean difference between tree nut and control interventions per one serving (28.4 g/day) over 3 to 26 weeks. The overall tree nut mean showed significantly lowered total cholesterol (TC) by −4.7 mg/dL (*p* = 0.001), LDL cholesterol (LDL-C) by −4.8 mg/dL (*p* = 0.01), with insignificant lowering of apolipoprotein B (Apo-B) by −3.7 mg/dL (*p* = 0.17), triglycerides (TG) by −2.2 mg/dL (*p* = 0.16), and HDL cholesterol (HDL-C) by −0.3 mg/dL (*p* = 0.33). The dose–response effects for nut intake and TC and LDL-C were non-linear (*p* < 0.001 each), with stronger overall effects observed for doses of ≥60 g tree nuts/day. In addition, there was a significantly stronger Apo-B lowering effect in T2 diabetic subjects than for healthy subjects by −9 mg/dL. The authors did not observe significant heterogeneity on TC and LDL-C across the different tree nuts when added to a variety of background diets. Significant effects on blood pressure (BP) and hs c-reactive protein (hs-CRP) were not identified. 

Musa-Veloso et al. [69], in a 2016 systematic review and meta-analysis of 18 RCTs with 27 datasets, evaluated the effects of almond-enriched diets on blood lipids in trials with a duration of ≥4 weeks. The mean main effect differences between almond and control diets on blood lipids levels showed that almonds significantly lowered TC, LDL-C, and TG, as summarized in Figure 16. The effect of almond serving size on blood lipids showed that intakes of ≥45 g almonds/day were significantly more effective in lowering TC and LDL-C than doses of <45 g/day, and neither dosage significantly changed HDL-C or TG levels, as summarized in Figure 17. Additionally, almonds were significantly more effective at lowering TC and LDL-C in individuals with elevated baseline lipid levels than those with normal healthy lipid levels.

Lee-Bravatti et al. [28], in a 2019 systematic review and meta-analysis, evaluated 15 RCTs (*n* = 534 healthy or at risk for CVD subjects) for the overall mean changes in blood lipids, fasting blood glucose, blood pressure (BP) and hs C-reactive protein (hs-CRP) between almond and control interventions. Almond diets significantly lowered TC by −11 mg/dL and LDL-C by −6.0 mg/dL, with significant heterogeneity (*p* < 0.01 for both) based on 13 trials. Almonds lowered HDL-C by −1.3 mg/dL with no heterogeneity (*p* < 0.29) based on 12 trials. Almonds tended to lower TG by 16 mg/dL at doses >42.5 g/day and by 29 mg/dL after durations >6 weeks. Fasting blood glucose was slightly but significantly reduced at almond doses >42.5 g/day by −4.1 mg/dL (*p* < 0.01), which is consistent with greater almond intake associated with gut bulking, which helps slow the rate of glucose absorption. Diastolic blood pressure (BP) mean from 7 trials was insignificantly lowered by 1.5 mm Hg, but there was a small reduction which was significant at almond dosages >42.5 g/day by −3.2 mm Hg and durations >6 weeks by −4.2 mm Hg (*p* < 0.01 for both). There were no significant changes in systolic BP at any almond dose or duration. The data on hs-CRP was too limited to run a subgroup analysis on. 

Eslampour et al. [70], in a 2020 systematic review and meta-analysis of RCTs (*n* = 16, 1128 subjects) on the effect of mean BP differences between almond and control interventions, confirmed the findings in Lee-Bravatti et al. [28]. Mean diastolic BP for the almond diets was significantly lowered by −1.30 mm Hg (−2.31, −0.30 mmHg; *p* = 0.01), with 0% heterogeneity. However, there was no significant lowering of mean systolic BP by −0.83 mm Hg (−2.55, 0.89; *p* = 0.34), with 59% heterogeneity. 

Ntzouvani et al. [71], in a 2019 systematic review of a small number of nut RCTs, indicated that the intake of 60 g/day of almonds by prediabetic subjects improved fasting blood glucose, fasting plasma insulin, HbA1c, insulin resistance (HOMA-IR), cellular glucose uptake by lymphocytes, and/or beta cell function compared to the control diet over 4 months. 

Viguiliouk et al. [72], in a 2014 meta-analysis of 12 RCTs in T2 diabetic subjects (*n* = 450), found that the median dose of 56 g of tree nuts/day significantly lowered mean hemoglobin A1c (HbA1c) by −0.07% (−0.10, −0.03 %; *p* = 0.0003) and fasting blood glucose by −2.7 mg/dL (−4.9, 0.4 mg/dL; *p* = 0.03), and tended to lower fasting blood insulin and HOMA-IR compared to control diets. 

Muley et al. [73], in a 2020 systematic review of 15 RCTs in T2 diabetes subjects, found that overall, tree nuts intake compared to controls lowered mean fasting blood glucose by −4.7 mg/dL and mean HbA1c by −0.11% over 3 months. 

#### 3.2.2. Individual RCTs on Almonds 

Healthy Individuals and Almond-Enriched Habitual Diets or Meals

All 15 RCTs showed that standardized meals or habitual diets with almonds (10−100/g/day) supported consistently large and significant beneficial effects on blood lipids over 4 to 24 weeks, but other metabolic health biomarkers such as blood pressure, inflammatory and oxidative stress markers, and endothelial function are usually clinically insignificant or statistically insignificant compared to control diets. Acute postprandial analysis for glycemic index or flow-mediated dilation shows limited impact on related longer time effects on glycemic, insulin, blood pressure, or endothelial health benefits. One reason for the lack of postprandial studies translating into longer-term benefits is that these are measured during one meal, usually breakfast, and these acute benefits are mitigated by subsequent meals, snacks, and beverage choices.

##### Acute RCTs

Josse et al. [74], in a parallel dose–response RCT, randomly assigned 9 healthy subjects (2 women and 7 men, mean age 28 years, and mean BMI 23) to consume 50 g of available carbohydrates from white bread with 0, 30, 60, or 90 g of almonds to assess postprandial glycemia with frequent blood glucose measures over 120 min. The combination of almonds with white bread progressively lowered the glycemic index of the meal in a dose-dependent manner compared to the white bread (control) (Figure 18). 

Jenkins et al. [75], in parallel RCTs, randomized 15 healthy subjects (7 men and 8 women, mean age 26 years, mean BMI 23) into one of two diet groups: (1) 97 g white bread (control) or (2) the control with 60 g of almonds, followed by frequent measures of postprandial blood glucose, serum insulin, and protein thiol (which is protective against protein oxidative damage) for 240 min. Compared to the white bread control group, the combination of almonds plus white bread reduced peak glucose by −14% (*p* < 0.001), insulin by −31% (*p* ≤ 0.042), and increased serum protein thiol levels by 15 mmol/L (*p* = 0.021). Adding almonds to parboiled rice and instant mashed potatoes had similar effects as shown for white bread. Almond’s effect of increasing postprandial protein thiol levels protects against reactive oxygen species (ROS) such as oxidized LDL-C and vascular endothelial protein damage to help reduce coronary heart disease (CHD) risk. 

Bhardwaj et al. [76], in a crossover, controlled feeding RCT, randomly assigned 27 overweight subjects (16 women and 11 men, mean age 42 years, mean BMI 29, normal blood pressure) into test diets of 1150 kcal meals consisting of sandwiches of white bread, salami, fatty cheese, full-fat yogurt, and water, plus walnuts (60 g) or almonds (76 g) to match the fat content. Vascular endothelial function measurements were taken over 4 h. Both almonds and walnuts similarly improved flow-mediated dilation (FMD) and decreased soluble vascular cellular adhesion molecules’ (sVCAM) function with no significant difference between the two nuts (*p* ≤ 0.51). Vascular endothelial dysfunction has been implicated in early and advanced atherosclerosis related to perfusion abnormalities and ischemic events. 

##### Four to Six Weeks Duration

Lovejoy et al. [32], in a parallel RCT, randomly assigned 20 healthy adults (10 men and 10 premenopausal women, mean age 25 years, mean BMI 23) to receive 100 g/day of whole almonds alone or mixed with a variety of foods (e.g., trail mix, cookies, or muffins) plus their habitual diet for 4 weeks. Almonds significantly decreased TC (−21%) and LDL-C (−29%) from baseline (*p* < 0.05 for both). TG was not significantly altered from baseline. HDL-C decreased significantly from baseline and was significantly lowered in men compared to women. Both TC:HDL-C and LDL-C:HDL-C ratios decreased with almond supplementation. In women, almonds tended to improve insulin sensitivity (*p* = 0.09) compared to baseline levels. 

Lee et al. [77], in a crossover, controlled feeding RCT, randomized 31 healthy overweight and obese individuals (13 women and 18 men, mean age 46 years, mean BMI 30) into isocaloric (2100 kcal/day) diets based on: (1) the average American control diet, (2) the control diet with 42.5 g raw almonds, (3) a chocolate diet with 18 g/day cocoa powder and 43 g/day dark chocolate, or (4) a combination of almonds and chocolate diets for 4 weeks. Compared with the control diet, the 42.5 g almond diet significantly lowered TC (*p* = 0.004), non–HDL-C (*p* = 0.006), and LDL-C (*p* = 0.003), with no significant changes in HDL-C or triglycerides (*p* > 0.05), as shown in Figure 19. Additionally, the almond diet significantly reduced large buoyant low-density lipoprotein particles by −5.4 mg/dL (*p* = 0.04). The combination chocolate and almond diet showed significantly lower amounts of small dense LDL particles by −6.7 mg/dL (*p* = 0.04). Measures of vascular health and oxidative stress were not significantly different between these diets.

Jung et al. [46], in a crossover RCT, randomly assigned 84 healthy overweight and obese Korean adults (87% women, mean age 52 years) to a 65% energy carbohydrate and 21% energy from fat Korean diet plus daily snacking on 56 g/day of roasted almonds or 79 g/day isocaloric home-made cookies for 4 weeks each. Snacking on almonds significantly decreased TC, LDL-C, and non-HDL-C by 5.5%, 4.6%, and 6.4% respectively, relative to the cookie control (*p* ≤ 0.05). Almonds also significantly reduced IL-10 compared to the cookie control by 22% (*p* < 0.04), and elevated levels have been associated with poor CVD prognosis [78]. There was a tendency to reduce inflammatory biomarkers intercellular adhesion molecule -1(ICAM-1), IL-1β, and IL-6 with almonds, compared to the cookies (*p* ≤ 0.187). No notable changes were observed in plasma ox-LDL, malondialdehyde, carbonyls, hs-CRP, tumor necrosis factor (TNF)-alpha, vascular cell adhesion molecule- (VCAM-1), or blood glucose status. Almonds increased plasma α-tocopherol by 8.5% (*p* ≤ 0.05) and decreased ỷ-tocopherol by 18% (*p* = 0.01) from the baseline. The almond diet improved the unsaturated - saturated fatty acid ratio and fiber intake while decreasing % energy intake from carbohydrate. The investigators concluded that snacking on almonds by overweight or obese Koreans in the typical Korean diet improves nutrition quality and reduces cardiovascular disease risk relative to a cookie snack.

Dikariyanto et al. [45], in a parallel RCT, randomized 107 healthy subjects (56 years, 70% women, mean BMI 27) into an average United Kingdom diet, including 20% of the daily energy intake from a snack of roasted almonds (63 g, ¼ cup) or an isocaloric control snack of mini-muffins for 6 weeks. Compared to the control sweet and savory muffin snack, almond snacking significantly reduced non-HDL-C and LDL-C, and increased flow-mediated dilation (FMD), and insignificantly reduced TC and HDL-C, and increased TG (Figure 20). Compared to the muffin snack, almond snacking improved heart rate functions of the parasympathetic nervous system during mental stress by optimizing vagal tone through better cardiac tissue responsivity to neurotransmitters and hormone changes [79]. There was no significant difference in fasting blood glucose, insulin, HOMA-insulin resistance, 24 h ambulatory or clinic blood pressure (BP), or liver fat between the groups. The almond diet improved diet quality by significantly reducing energy % intake from total carbohydrates, sugar, and starch, and improving the unsaturated–saturated fat ratio by 1.3 (*p* < 0.001), and significantly lowering daily sodium by −671 mg, and increasing fiber by 7.4 g, vitamin E by 15.4 mg, magnesium by 149 mg, and potassium by 574 mg. The authors concluded that the level of improvement in FMD and LDL-C by eating almonds instead of typically consumed high-carbohydrate snacks has the potential to reduce CVD risk by up to 30%. 

##### Eight to Twenty-Four Weeks Duration

Dhillon et al. [44], in a parallel RCT, randomized 73 healthy college students (mean age 18 years, 58% women, mean BMI 26) into a roasted almond snack group (57 g/day; 364 kcal) or Graham cracker control group (78 g/day; 338 kcal/day) for 8 weeks to evaluate their effects on metabolic health outcomes. Compared to the cracker group, almond snacking resulted in a significantly lower drop of HDL-C by 11%, 13% lower 2 h glucose area under the curve, 34% lower insulin resistance index, and 82% higher Matsuda insulin sensitivity index during the oral glucose tolerance test (OGTT), despite similar changes in body mass compared to crackers (*p* > 0.05). Both almond and cracker snacking had similar reduced fasting blood glucose and insulin and LDL-C, whereas glucagon-like peptide-1 (GLP-1), which improves insulin secretion by the pancreas, significantly increased from baseline to week 4, but then significantly decreased between week 4 and week 8 for no significant net change over 8 weeks. In general, almonds significantly improved HOMA-β, an index of fasting β-cell function (*p* < 0.05). Almond snacking by breakfast-skipping freshman college students had some beneficial effects on cardiometabolic and glucoregulatory health. 

Liu et al. [42], in a parallel RCT, had 169 healthy adults (54% women, mean age 26 years, and mean BMI 23) continue their habitual diets with the avoidance of any nuts or nut products before randomly assigning them to one of three supplemental food regimens: (1) 56 g (320 kcals) of almonds divided into 3 portions with 18 g almonds as a pre-meal before each meal, (2) 56 g of almonds for snacking between meals, or (3) an isocaloric cookie snack (control) for 8 and 16 weeks. The changes in blood lipids from baseline to weeks 8 and 16 resulting from pre-meal almonds, almond snacks, and the control cookie snack are summarized in Figure 21A,B [42]. Compared to the control cookies, snacking on almonds significantly reduced TC (*p* = 0.043), LDL-C (*p* = 0.011), and non-HDL-C (*p* = 0.013) at 8 and 16 weeks. In addition, compared to pre-meal almonds and control cookies, almond snacking significantly reduced LDL-C levels at week 16. Changes in triglyceride and HDL-C levels were not significantly different between the three groups. The changes in oxidative or inflammatory biomarkers did not significantly differ among the groups. 

Liu et al. [43] continued the previous trial [41] for 4 additional weeks to 20 weeks with 57 subjects from the almond (56 g) snack group (54% men, mean age 29 years, and BMI 23) and 28 cookie subjects (50% men, mean age 26 years, and BMI 22). Compared to the cookie control, almond changes from baseline to 20 weeks significantly lowered LDL-C by 7.0 mg/dL, TC by 10 mg/dL, non-HDL-C by 6.0 mg/dL, VLDL-C by 3.4 mg/dL, and TG by 15 mg/dL (*p* < 0.0001). Almonds had a small but significant diastolic BP-lowering trend over time compared to the cookie control (Figure 22). 

Jamshed et al. [37], in a parallel RCT, randomly assigned 150 coronary artery disease patients with controlled LDL-C and low HDL-C (74% men, mean age 60 years, overweight, mean low HDL-C ≤ 36 mg/dL) into their habitual diet with no almond control or 10 g/day American or Pakistani almonds that were soaked overnight and consumed before breakfast for 12 weeks. Almonds significantly increased HDL-C by an average of 10% after 6 weeks and 14% after 12 weeks compared to baseline (*p* < 0.05). Additionally, almonds significantly reduced TC, LDL-C, VLDL-C, TG, and TC:HDL-C and LDL-C:HDL-C ratios at both weeks 6 and 12 (*p* < 0.05). The atherogenic index was significantly reduced in the almond diet compared to baseline at both 6 and 12 weeks (Figure 23).

Coates et al. [49], in a parallel RCT, randomized 151 overweight and obese individuals (mean age 65 years, mean BMI 30) into 15% of energy as snacks from raw almonds or a nut-free habitual diet with carbohydrate-rich biscuits or potato chips along with substitution dietary counseling to maintain stable weight for 12 weeks. Compared to the high-carbohydrate snacks, the almond snacks significantly reduced TG by −14.2 mg/dL (*p* = 0.008) and systolic BP by −4.0 mmHg (*p* ≤ 0.044), with no other significant differences in other cardiometabolic biomarkers. There were no differences between the diets on mood or cognitive performance. These investigators currently have a 9-month RCT in progress for almonds vs. carbohydrate-rich control to comprehensively assess weight and body composition, appetite regulation, cardiometabolic health, liver function, inflammatory biomarkers, mood, pain, and quality of life [80].

Rakic et al. [81], in a parallel, single blinded RCT, randomized 68 “healthy” middle-age/older adults (mean age 62 (50–70) years, mean BMI 29, about 2/3 graduated from college, and Caucasians) into one of three snack groups: 42 g almonds, 84 g almonds, or a 100 g cereal snack of coconut, butter, and beef jerky mixture (control) for 6 months. Serum tocopherol, oxidative and inflammation status, and cognitive performance were measured at baseline, three months, and six months. After 6 months, α-tocopherol was significantly increased by 8% for the 84 g almond diet compared to baseline (*p* < 0.05), but α-tocopherol was not increased in the other groups. The 42 g almond group had a significant improvement in the Motor Screening Task, which is a general assessment of sensorimotor function and comprehension, at 3 months but not at 6 months. The 84 g almond group at 6 months showed significant improvements in visuospatial working memory (*p* = 0.023), in visual memory and learning (*p* = 0.017), and in spatial planning and working memory (*p* = 0.001), and the control snack group had no significant improvement in the functions at any time (Figure 24). However, changes in any cognitive scores over time did not statistically differ among the three groups. Thus, the independent effect of almonds on cognitive performance may be relatively small, and studies for longer duration, with a larger number of subjects and other control diets, are warranted to provide additional evidence of the beneficial effects of almonds on cognitive performance in middle-aged and older adults. The control snack mix group significantly increased TC by about 4.5% after 3 and 6 months (*p* ≤ 0.048), but there were no significant changes in either of the almond groups. Serum biomarkers of inflammation (e.g., hs-CRP, IL-6) and oxidative stress (e.g., red blood cell glutathione peroxidase) were not significantly altered in any of the diets. The consumption of 84 g (3 serving)/day over 6 months showed encouraging improvements in cognitive performance in middle-aged and older adults after 6 months. 

Foolad et al. [82], in a parallel, investigator-blinded RCT, randomized 31 healthy postmenopausal women (mean age 62 years, mean BMI 30) into habitual diets with 60 g of almonds (20% of energy) or isocaloric carbohydrate-based snacks, including cereal, granola bars, or pretzels, for 16 weeks. With high-resolution photographic analysis, the almond group significantly decreased in % change in wrinkle severity and width compared to the carbohydrate-rich control group (Figure 25). However, there were no significant differences in sebum production or trans-epidermal water loss between the almond and control groups.

Li et al. [83], in a parallel RCT, randomly assigned 39 Asian women (mean age 28 years, mean BMI 23) to one of two habitual diets, with 42.5 g (246 kcals) almonds or 54 g pretzels (200 kcals), for 12 weeks. This study showed that daily almond intake enhanced protection from UVB photo-damage (the shorter wavelength associated with skin burning) by stimulating increased minimal erythema dose (amount of radiation required to produce skin redness).

Hyperlipidemic Individuals and Almond-Enriched Heart-Healthy Diets

Ten of eleven RCTs in hyperlipidemic, CHD, or abdominally obese individuals found that almond intake (34–100/g) along with heart-healthy diets improved cardiometabolic biomarkers or reduced 10-year CHD risk. 

Sabate et al. [50], in a dose–response, crossover RCT, randomized 25 mildly hyperlipidemic adults (14 men and 11 women, mean age 41 years, and BMI > 30) into one of three isocaloric (2400 kcal) NCEP Step 1 diets: (1) 0 almond control, (2) low almond 10% energy (34 g/day), or (3) high almond 20% energy (68 g/day), for 4 weeks. The dose–response relationships between the percentage of energy from almonds in the Step 1 diet and changes in TC, LDL-C, Apo-B, and in HDL-C are shown in Figure 26.

Rajaram et al. [84], in a crossover, controlled RCT, randomly assigned 25 healthy adults (14 women and 11 men, mean age 41 years, mean body weight 71 kg) to one of three heart-healthy 2000 kcal diets for 4 weeks each: (1) nut-free control diet (<30% energy from fat), (2) low almond diet (10% of energy, 34 g almonds/day, 35% total energy from fat), or (3) high almond diet (20% of energy, 68 g almonds/day, 39% total energy from fat), consumed as snacks or incorporated into foods such as pizza or salads. Serum E-selectin (a cell adhesion molecule expressed by endothelial cells) decreased as the % energy from almonds increased (*p* < 0.0001). C-reactive protein (hs-CRP) in both almond diets was significantly lowered compared to the control diet (*p* < 0.03). There were no diet-related changes in IL-6 or fibrinogen. This study provides evidence that incorporating 68 g almonds/day into the diet supports improved vascular health beyond lowering LDL-C.

Nishi et al. [85], in a crossover dose–response RCT, randomized 27 hyperlipidemic subjects (15 men and 12 postmenopausal women, mean age 64 years, mean BMI 26) to one of three isoenergetic NCEP Step 2 diets supplemented with mean 420 kcal/day: (1) full-dose almonds (73 g/day), (2) half-dose almonds with half-dose muffins, and (3) full-dose muffins for 4 weeks. Both almond-supplemented diets increased the oleic acid - saturated fatty acid ratio of the serum triglyceride fatty acids and non-esterified fatty acids compared to the muffin control diet. Every 30 g/day increase in almond intake was associated with a decrease in the estimated 10-year CHD risk score by 3.5 % (R = −0.247, *p* = 0.026, Figure 27).

Spiller et al. [57], in a parallel RCT, randomly assigned 38 hyperlipidemic subjects (12 men and 26 postmenopausal women, mean age 61 years, >25 BMI) into heart-healthy diets including 100 g of raw or roasted almonds or almond butter for 4 weeks. Raw almonds, roasted almonds, and almond butter had similar reductions in TC and LDL-C from baseline: raw almonds decreased TC by 17 mg/dL (*p* < 0.01) and LDL-C by 19 mg/dL (*p* < 0.002), roasted almonds decreased TC by 12 mg/dL (*p* < 0.034) and LDL-C by 11 mg/dL (*p* < 0.012), and almond butter reduced TC by 11 mg/dL and LDL-C by 11 mg/dL (*p* < 0.034). No significant changes in TG, VLDL-C, or HDL-C were shown for the whole almonds, but almond butter tended to increase HDL-C by 5 mg/dL and decrease TG by 23 mg/dL compared to no change for raw and roasted almonds from baseline. 

Damasceno et al. [60], in a crossover RCT, entered 18 healthy, hypercholesterolemic adults (mean age 56 years, mean BMI 26) into a general Mediterranean diet with limited guidance for the run-in (6.9% energy from saturated fat) for 4 weeks, before randomly assigning them to one of three healthier Mediterranean diets with 40% of baseline fat replaced with: (1) virgin olive oil (35–50 g/day), (2) walnuts (40–65 g/day), or (3) almonds (50–75 g/day), with a mean of about 5% energy from saturated fat for 4 weeks each. All three healthy fat sources similarly significantly improved TC, LDL-C, and LDL-C:HDL-C from baseline (*p* < 0.012 for all). There were no improvements in HDL-C, TG, systolic BP, homocysteine, oxidized LDL-C, hs-CRP, or cell adhesion molecules from baseline.

Berryman et al. [53], in a crossover, controlled feeding study, randomized 48 healthy hyperlipidemic subjects (26 women and 22 men, mean age 50 years, mean BMI 26) into identical cholesterol-lowering nut-free diets augmented with 42.5 g of raw almonds/day or an isocaloric high-carbohydrate muffin for six weeks. Compared to the muffin control diet, the almond diet significantly decreased TC, LDL-C, non-HDL-C, very-low-density lipoproteins (VLDL), Apo-B, and HDL-C (Figure 28). In addition, compared to the control diet, the almond diet lowered TG by –7.2 ± 6.0 mg/dl (*p* = 0.24), hs-CRP by −0.34 ± 0.18 mg/L (*p* = 0.03), and fasting glucose by −0.7 ± 1.5 mg/dL (*p* = 0.61). The substitution of almonds for high-carbohydrate snacks appears to be a simple approach to prevent the onset of cardiometabolic diseases in at-risk healthy individuals.

Ruisinger et al. [58], in a parallel RCT, randomly assigned 48 hyperlipidemic subjects with ongoing statin therapy (50% women, mean age 60 years, mean BMI 29) into the NCEP Step 3 diet program with 100 g of almonds or a standard Step 3 diet program (control) for 4 weeks. Compared to the control Step 3 diet, the almond-enriched Step 3 diet significantly lowered non-HDL-C, very-low-density lipoproteins (VLDL-C), intermediate density lipoproteins cholesterol (IDL-C), and tended to reduce total cholesterol, LDL-C, triglycerides, and lipoprotein (a) without significantly lowering HDL-C compared to the control diet (Table 5). There was also a shift from LDL pattern A to type B pattern particles in the almond group (*p* = 0.003). The investigators suggested that almonds may be used as adjunctive therapy with statins to further reduce CHD risk.

Jenkins et al. [51], in a dose–response crossover RCT, randomized 27 hyperlipidemic subjects (15 men and 12 postmenopausal women, mean age 64 years, mean BMI 26) to three isoenergetic NCEP Step 2 diets with 420 kcals/day supplements consisting of: (1) full-dose almonds (73 g/day), (2) half-dose almonds (36.5 g/day) with half-dose muffins, and (3) full-dose whole-wheat flour muffins (control) for 4 weeks. The control muffin diet significantly raised TG by 10.8% ± 4.7% from baseline, with no other significant changes in lipoproteins. Compared to baseline, almond diets reduced blood lipid concentrations in a dose–response manner (Figure 29). Compared to the muffin control, both almond doses significantly reduced oxidized LDL-C from baseline and the full dose of almonds significantly reduced the estimated 10-year risk of CHD from baseline and control diet (Figure 30). There was a 1% reduction of LDL-C for each 7 g intake of almonds. 

Chen et al. [86], in a crossover RCT, randomized 45 coronary artery disease (CAD) patients with elevated blood lipids (69% women, mean age 62 years, mean BMI 30; stable medication usage) into a control NCEP Step 1 diet or a NCEP Step 1 diet with 85 g of almonds/day for 6 weeks. All the subjects were receiving polypharmacy therapies and ongoing active medical care. Compared to the control diet, the almond diet did not significantly change the serum lipid profile, blood pressure (BP), or systemic inflammatory biomarkers. The almond-rich Step 1 diet increased the intake of the amino acid arginine, a biological precursor of nitric oxide to improve blood flow, by 62% compared to the control Step 1 diet. Compared to the control diet, the almond diet tended to decrease circulating vascular cell adhesion molecule-1 by −7.8% (*p* = 0.064), increase urinary nitric oxide by 17.5% (*p* = 0.09), and improve the ratio of α to gamma tocopherol (Figure 31). 

Berryman et al. [87], in a crossover, controlled feeding RCT, randomized 48 middle-aged, overweight individuals with elevated LDL-C (26 women and 22 men, mean age 50 years, mean BMI 26) to a heart-healthy diet with 42.5 g of raw almonds/day or an isocaloric banana muffin plus 2.7 g of butter for 6 weeks. This study found that substituting almonds for a carbohydrate-rich snack in lower saturated fat diets improved plasma HDL-C subspecies and cholesterol efflux from arterial circulation in normal weight individuals with elevated LDL-cholesterol to lower ischemic heart disease (IHD) risk. 

Williams et al. [88], in a crossover RCT, randomized 24 abdominally obese adults (9 men and 15 women, mean age 46 years, mean BMI 31) to healthy high-fiber (mean 34 g/day) and low saturated fat (mean 7% energy) isocaloric diets, with: (1) a higher 50% carbohydrate energy reference diet, (2) a 50% carbohydrate energy diet with 20% substitution of the energy by raw almonds (50% from whole almonds and 50% from almond meal added to spreads, soups, and cakes), or (3) a 25% carbohydrate energy reference diet for 3 weeks. All diets included healthy fat sources such as combinations of avocados, olives, or almonds, and fruits and dairy were a primary source of sugar, with dietary consulting and monitoring by a staff nutritionist. Compared to the 25% carbohydrate energy diet, the 50% carbohydrate diet with almonds and the 50% carbohydrate control had similar significantly increased small, dense LDL-C concentration, by 28.6 ± 10.4 nmol (*p* = 0.008) and reduced LDL-peak diameter by −1.7 ± 0.6 units (*p* = 0.008). There were no other significant differences in plasma LDL-C or other lipoproteins, fasting glucose or insulin, HOMA-IR, or blood pressure for the 50% carbohydrate energy diets with almonds or without almonds. Individuals with abdominal obesity, consuming the 50% carbohydrate energy diet with 20% of energy from a combination of whole almonds and almond meal, were not provided any extra protection from atherogenic small, dense LDL-C compared to a 50% carbohydrate control diet.

Type 2 (T2) Diabetic or Prediabetic Individuals and Almond-Enriched Diets

Twelve RCTs in T2 diabetic or prediabetic individuals consuming 28 to 85 g/day of almonds in habitual diets and healthy diets (e.g., NCEP, Diabetes Associations diets, low-carbohydrate diets) for 3 to 24 weeks showed variable levels of improved glycemic control and CVD risk biomarkers compared to control diets [34,39,46,47,50,55,56,57,62,88,89].

##### Habitual Meals or Diets

Bodnaruc et al. [90], in a crossover RCT, randomized 7 men with T2 diabetes on a Metformin regimen (mean age 64 years, mean BMI 29) to one of two experimental breakfast meals (710 kcals): (1) white bread (97 g) and roasted almonds (85 g) as the intervention breakfast and (2) white bread (115 g), unsalted butter (24 g), and cheddar cheese (69 g) as the control, before switching diets. Blood samples were taken at 0 (fasting), and 15, 30, 60 90, 120, and 240 min postprandially for glycemic and hormonal changes. The almond breakfast significantly improved postprandial glycemia, insulinemia, and estimated glucose metabolic clearance rate (a measure of postprandial insulin sensitivity) and tended to increase glucagon-like peptide-1 (GLP-1, a hormone that regulates appetite and blood glucose levels) (Figure 32). Almonds had an adjunctive beneficial impact on glycemic and hormonal responses in men with T2 diabetes on Metformin regimen. 

Bowen et al. [47], in a parallel RCT, randomly assigned 76 adults with elevated risk of or with T2 diabetes (61 years, 59% male, mean BMI 34) into their habitual diet plus the replacement of their usual snacks with 1 of 2 snacks divided into two servings/day: (1) raw almonds (56 g/day) or (2) an isocaloric sweet biscuit (control) snack for 8 weeks. Compared to the control snack, the almond snack significantly reduced TC:HDL-C ratio in women by −5.0% (*p* = 0.05) compared to –1.8% in men (*p* = 0.46) and −2.5% for all subjects (*p* = 0.15). Compared to the biscuit snack, the almond snack tended to reduce TC by 3.1% (*p* = 0.08), LDL-C by −4.8% (*p* = 0.07), and hs-CRP by 42% (*p* = 0.12). There were no differences between the groups for HDL-C, TG, plasma glucose, serum insulin, or HbA1c.

Palacios et al. [48], in a crossover RCT, randomized 33 adults with prediabetes (48% women, mean age 48 years, mean BMI 31) into their usual diet plus snacks consisting of 84 g of raw almonds/day (240 kcals, 1.5 (42.5 g) servings twice daily) or an isocaloric variety of carbohydrate snacks including baked chips, pretzels, dried mangos, pudding, cookies, mini bagels, or rice cakes for 6 weeks. The subjects were instructed to use food substitution in their usual diet during each diet arm to maintain baseline energy intake. In this relatively healthy prediabetes cohort, the mean baseline measures for fasting glucose, insulin, lipoproteins, TG, hs-CRP, and blood pressure were generally within the normal range. After 6 weeks, there were no significant differences in baseline metabolic health markers between the almond and carbohydrate-rich snacks, except that subjects with the almond snack diet had significantly fewer small, denser LDL_3 + 4_-C particle fractions by 12.2 mg/dL (*p* = 0.024). These small, dense LDL-C particles are selectively attracted to the arterial cell wall and stimulate pro-oxidative inflammatory promoters of atherogenesis [89]. 

Sweazea et al. [34], in a parallel RCT, randomized 21 adults with T2 diabetes (57% women, mean age 56 years, mean BMI 35) into one of two groups for 12 weeks. The control group was instructed to reduce nut intake to ≤2 servings of non-almond nuts weekly and otherwise maintain the usual diet. The almonds group was instructed to add 42.5 g of almonds 5–7 days/week to their usual diet. The almond snack significantly lowered hs-CRP compared to the control (Figure 33) and tended to increase HDL-C by 2.7 mg/dL (*p* = 0.105), but there were no other significant changes in blood lipids, glucose regulatory, or inflammatory biomarkers. 

Cohen and Johnston [39] conducted two RCTs: (1) A pilot parallel RCT randomly assigned 13 obese subjects with T2 diabetes (54% men, mean age 66 years, mean BMI 35) to an isocaloric diet supplemented with one serving of almonds (28 g/day, approximately 160 kcals) or two cheese sticks (160 kcals) for 5 days/week for over 12 weeks. The almond snack significantly lowered HbA1c by −5.0% compared to the cheese snack. (2) In an acute RCT, 12 non-diabetic and 7 diabetic individuals consumed a standardized dinner (sub sandwich, chips, and soda) followed by a 12 h overnight fast, after which they were assigned to an isocaloric early morning breakfast consisting of a white bagel with butter and berry juice (control) or the same meal with 28 g of almonds replacing butter. Blood draws were taken at 0, 30, 60, 90, and 120 min. In T2 diabetic individuals, the almond diet significantly reduced postprandial glycemia with −30% lower AUC over 120 min (*p* = 0.043) and reduced peak glucose level by −0% (*p* = 0.046) compared to the control diet. GLP-1, which helps to activate pancreatic beta-cell insulin secretion, at 30 min post-meal, was insignificantly increased by 28% vs. the control diet. These two small trials provide support for almonds as a healthy snack for individuals with T2 diabetes.

##### Healthy Diets

Gulati et al. [56] conducted a pre–post-clinical trial in which 50 compliant subjects with T2 diabetes from New Delhi, India (54% men, mean age 46 years, mean BMI 29), were enrolled for a 3-week run-in diet based on the dietary guidelines for Asian Indians with T2 diabetes, followed by a post-run-in diet which included 20% energy from raw almonds (about 60 g/day) incorporated by substituting the almonds for visible fatty foods and a portion of carbohydrates from the run-in with guidance to maintain isocaloric energy with the run-in diet for 24 weeks. Compared to the run-in diet, the almond-enriched diet significantly improved TC, LDL-C, VLDL-C, triglycerides, HbA1c, and hs-CRP, and tended to improve pulse wave velocity (Table 6). 

Li et al. [54], in a crossover, controlled feeding RCT, randomized 20 T2 diabetic individuals with hypoglycemic medication, but not insulin, and borderline high LDL-C (55% postmenopausal women, mean age 58 years, mean BMI 26) into 1800 kcal diets: a nut-free control Chinese cuisine prepared following the NCEP Step 2 diet guidelines or an isocaloric roasted almond-supplemented Step 2 diet to replace 20% of daily energy intake (average 56 g almonds/day) diet for 4 weeks. Compared to the control diet, the almond diet significantly lowered TC, LDL-C, Apo-B, and LDL-C/HDL-C ratio, non-esterified fatty acids (NEFA), and homeostasis model assessment of insulin resistance index (HOMA-IR) (Figure 34). Additionally, the almond diet significantly reduced fasting insulin by −4% and glucose by −0.8% (*p* ≤ 0.238) and increased plasma α-tocopherol by 23% (*p* ≤ 0.0001). The inclusion of almonds into a healthy diet improved glycemic regulation and decreased cardiovascular disease risk factors in individuals with T2 diabetes. 

Wien et al. [62], in a parallel RCT, randomized 65 adults with prediabetes (74% women, mean age 54 years, mean BMI 30) into the American Diabetes Association (ADA) diet containing 20% of energy from roasted almonds (60 g/day with instructions to avoid other nuts) or a nut-free ADA diet for 16 weeks, with both diets including individualized dietitian counseling (e.g., exchanging added almonds for other foods or snacks), with a prescribed target energy of 250 kcals/day deficit in subjects with BMI > 25 kg/m^2^ (51 out of 65 subjects). Compared to the control diet, the almond-enriched diet significantly improved biomarkers of insulin sensitivity and lowered LDL-C (Figure 35). Additionally, the almond diet increased vitamin E intake by 150% and fiber intake by 22% and reduced saturated fat by 10% compared to the control diet. The researchers suggested that the inclusion of almonds into the ADA diet could help to reverse prediabetes status or prevent the transition from prediabetes to T2 diabetes. 

Richmond et al. [50], in a crossover, controlled feeding RCT, randomly assigned 22 postmenopausal women with T2 diabetes (mean age 62 years, mean BMI 29) to a recommended dietary pattern for T2 diabetes plus 30 g of almonds/day or sunflower seeds (one serving) for 3 weeks, with a 4-week washout between groups. Compared to sunflower seeds, almonds resulted in a significantly smaller reduction of HDL-C (*p* = 0.05), a significant increase in α-tocopherol (*p* = 0.013), and a significantly smaller reduction in TG (*p* < 0.001). Both almonds and sunflower seeds significantly reduced TC, LDL-C, ox-LDL, and diastolic BP compared to baseline (*p* ≤ 0.02). Systolic BP was only significantly lowered by sunflower seeds, by 5.1% (*p* = 0.016), compared to 3.9% for almonds (*p* > 0.05). For glycemic control, almonds lowered blood glucose by 10% (*p* = 0.006) and sunflower seeds lowered blood glucose by 11% (*p* = 0.002). For insulin, there were no differences between almonds and sunflower seeds. Postmenopausal women with T2 diabetes including a daily serving of almonds or sunflower seeds had improved CVD and glycemic regulation biomarkers compared to baseline values. 

Chen et al. [56], in a crossover RCT with controlled feeding, randomized 33 patients with T2 diabetes (61% women, mean age 55 years, mean BMI 25) into a NCEP Step 2 diet control or an isocaloric replacement of 20% of the energy with almonds (60 g) for 12 weeks. Compared to the control diet, 27 subjects with the baseline HbA1c ≤ 8 when on the almond-enriched diet significantly lowered fasting serum glucose by −5.9% (*p* = 0.01) and HbA1c by −3.0% (*p* = 0.04). There were no significant changes between the diets for TC or LDL-C blood levels, which remained below 200 and 105 md/dL over the course of the study, respectively. Additionally, there were no differences between the two diets for HDL-C, TG, or Apo-B levels. The almond diet contained significantly more fiber, alpha-tocopherol, magnesium, and less % energy from carbohydrate, with no significant differences in total energy intake.

Ren et al. [59], in a parallel RCT, randomly assigned 45 subjects with T2 diabetes (44% men, mean age 72 years, mean BMI 23) to one of two isocaloric diets: (1) an almond-based low-carbohydrate diet (56 g almonds replacing 150 g of carbohydrate foods) or (2) a control low-fat (25% energy from fat) diet for 12 weeks with dietary counseling. Compared to the low-fat diet, the almond low-carbohydrate diet significantly reduced HbA1c % and the depression score (Figure 36).

Hou et al. [91], in a parallel RCT, randomly assigned 25 adults with T2 diabetes (15 men and 10 women, mean age 69 years, mean BMI 23) into low-carbohydrate diets with part of the starchy foods replaced by peanuts (60 g for men and 50 g for women) or almonds (55 g for men and 45 g for women) for 12 weeks. Compared to baseline, fasting blood glucose and post-prandial 2 h blood glucose decreased similarly in both the almond and peanut diets (*p* > 0.05), but the almond diet showed significantly lowered HbA1c by −0.58% (*p* = 0.043) vs. a −0.05% change for peanuts. There were no other significant differences in other metabolic health outcomes, including blood lipids or inflammatory biomarker IL-6, between peanuts and almonds. 

Almond Weight Loss Diets and Metabolic Health Biomarkers

Five LCD RCTs showed that 250–500 kcal deficit almond diets improved metabolic health biomarkers, similar to a nut-free control, but 1000 kcal deficit almond diets significantly improved these biomarkers compared to nut-free diets [63,64,65,66,67]. 

Foster et al. [63], in a parallel RCT, randomized 123 overweight and obese subjects (90% females, mean age 47 years, BMI 34) to 1 of 2 low-calorie diets (LCDs): (1) enriched with raw and roasted almonds (56 g/day consumed in two servings) or (2) a nut-free LCD. The targeted energy deficit was approximately 500 kcal/day (1200–1500 kcal/day for women and 1500–1800 kcal/day for men), with standard behavioral methods of hypocaloric weight loss for 18 months. Compared to the nut-free LCD, the almond LCD significantly lowered TG, TC, and TC:HDL-C, and tended to lower VLDL-C after 6 months, but there was no significant difference after 18 months (Table 7). There were no significant changes in resting systolic or diastolic BP between the two diets at either 6 or 18 months. 

Dhillon, Tan, and Mattes [64], in a parallel RCT, randomized 86 healthy adults (70% women, mean age 34 years, mean BMI 30) to 1 of 2 LCDs (targeted 500 kcal deficit/day) with roasted almonds 15% of energy (38 g/day) or a nut-free control for 12 weeks. Compared to the nut-free LCD, the almond LCD diet significantly lowered resting diastolic BP by –3.6 mmHg in the compliant group (*p* = 0.029), whereas the resting systolic BP was significantly lowered in both diets by 3.2 mmHg for the almond diet and −2.2 mmHg for the control diet from baseline (*p* < 0.05). Serum insulin and glucose, TG, TC, LDL-C, and HDL-C were statically unchanged over the course of this trial. Since half of the adults in the United States have hypertension, the consumption of LCD diets with almonds might be an option to help reduce hypertension in adults. 

Wien et al. [65], in a parallel weight loss RCT, randomized 65 obese and metabolic syndrome middle-aged adults (57% women, mean age 55 years, mean BMI 38) into either a nut-free, complex carb (rich in starchy vegetables and pasta) control LCD 1000 kcal/day or an almond-enriched LCD (1000 calorie/day) including 84 g of almond/day for 6 months under free-living conditions. The almond diet contained 39% energy from fat and 32% energy from carbohydrates vs. the complex carbohydrate diet with 18% energy from fat and 53% energy from carbohydrates. Compared to the control LCD, the almond LCD significantly decreased HDL-C and systolic BP, and both diets significantly lowered fasting glucose and insulin, diastolic blood pressure, and HOMA-IR (Figure 37). TC, TG, LDL-C, and LDL-C to HDL-C ratio were significantly decreased to a similar extent by both LCDs. Compared to the complex carbohydrate-based LCD, the almond-enriched LCD more effectively improved metabolic syndrome risk biomarkers. 

Abazarfard et al. [66], in a parallel weight loss (1000 kcals deficit) RCT, randomized 108 overweight and obese Iranian women (mean BMI 30, mean age 43 years, 93% completers) into two balanced LCDs for 12 weeks: (1) 50 g almonds/day divided into two servings or (2) a nut-free group. Compared to the nut-free LCD, the almond-enriched LCD significantly reduced triglycerides, TC, TC:HDL-C, fasting blood glucose, and diastolic BP, but not LDL-C or systolic BP (Figure 38). 

Abazarfard et al. [67], in a parallel weight loss (1000 kcal deficit) RCT, randomized 108 overweight and obese Iranian women (mean BMI 30, mean age 43 years, 93% completers) into a two-arm balanced LCD with 50 g almonds/day divided into two servings or a nut-free group for 12 weeks. Compared to the nut-free LCD, the almond-enriched LCD significantly lowered liver enzymes in obese women by −4.9 units/liter for alanine aminotransferase (ALT), 39 units/liter for aspartate aminotransferase (AST), and –5.2 units/liter for gamma-glutamyltransferase (GGT) *p* = <0.001 for all. The researcher suggested that these liver enzymes lowering effects might be related to almonds being a good source of vitamin E and magnesium. 

### 3.3. Colonic/Fecal Microbiota Biomarkers

#### 3.3.1. Systematic Reviews and Meta-Analyses of Nuts

A 2020 systematic review [92] and meta-analysis [93] of RCTs assessed the effects of nuts on fecal microbiota. However, the number of trials in these reviews is relatively small, with only 8 RCTs in each analysis. These analyses indicated that increased nut intake including almonds had the capacity to enhance fecal microbiota diversity and overall health biomarker profiles, but most current trials are limited or consist of relatively weak exploratory studies. Nut microbiota research is currently at a relatively early stage of understanding as there are wide variations in quality of trial design, methods, age and health status of subjects, level, type, and duration of nut intake and baseline dietary pattern. 

#### 3.3.2. Individual RCTs on Almond-Enriched Diets

Presently, eight RCTs show early evidence that whole or chopped almonds support a healthy microbiota by promoting microflora richness and diversity, increased symbiotic bacteria and reduced pathogenic bacteria, improved fiber metabolism by increasing butyrate and other SCFA-producing bacteria, and reduced fecal pH [59,94,95,96,97,98,99,100].

Choo et al. [94], in a parallel RCT, randomized 69 adults (mean age 61 years, mean BMI 33) to 2 servings (56 g) of whole raw almonds/day or an isocaloric biscuit snack for 8 weeks. The almond snack group had significant changes in microbiota composition (*p* = 0.011) and increases in bacterial richness, evenness, and diversity (*p* < 0.01) compared to the biscuit snack group. Almond snacking principally increased both the relative and absolute abundance of the symbiotic Ruminococcaceae family (*p* = 0.002), which has been linked to improved regulation of glycemic control and body weight. The almond snack showed potentially decreased obesogenic effects associated with changes in trends in fecal short-chain fatty acids (SFCA) for acetate (−8.4%, *p* = 0.13), propionate (+6.5%, *p* = 0.35), and butyrate (+5.1%, *p* = 0.36) from baseline, compared to less robust changes in the biscuit snack for acetate (−3.6%, *p* = 0.73), propionate (−0.8%, *p* = 0.47), and butyrate (+6.3%, *p* = 0.50) [93]. Although these changes in colonic SCFAs were not statistically significantly changed in the colon, a review article by Magne et al. [101] provides insights that their increased absorption into circulation could have clinically important effects on reduced risk of weight gain or obesity, as follows: (1) decreased circulatory levels of acetate leads to slowed “de novo” hepatic synthesis of lipids and ghrelin associated with decreased fat storage and appetite, (2) increased circulatory propionate stimulates appetite hormones such as GLP-1 and Peptide YY released by L-entero-endocrine cells to reduce appetite and “de novo” synthesis of fatty acids in the liver associated with less fat storage, and (3) increased circulatory levels of butyrate are associated with the regulation of energy metabolism and increased leptin gene expression along with increased insulin sensitivity and anti-inflammatory activities. In conjunction with SCFA changes, the almond snack significantly lowered fecal pH to 7.0 pH (*p* = 0.03) compared to the biscuit snack, maintaining a pH of 7.4. The lower pH is generally a signal of a healthier colonic microbiota [92]. In addition, regular almond consumption increased the abundance of beneficial symbiotic ruminococci, in the colonic microbiota in individuals with elevated blood glucose. 

Ukhanova et al. [95], in a crossover, controlled-feeding RCT, fed 18 healthy adults (10 men and 8 women, mean age 56 years, mean BMI 27) whole raw almond daily doses of 0, 42, or 84 g for 18 days. The data were generally presented in a qualitative manner. Almond consumption led to a modest increase in microbiota bacteria beta-diversity, improved efficiency of fiber metabolism, and increased butyrate-producing bacteria. Each are potentially important signals of a healthy colonic microbiota.

Holscher et al. [96], in a crossover, controlled-feeding RCT, randomized 18 healthy adults (10 men and 8 women, mean age 57 years, mean BMI 30) into one of five diets: (1) 0 g control, or 42 g/day of the following almonds: (2) whole raw, (3) whole roasted, (4) roasted, chopped, and (5) almond butter for 18 days. Although almond intake did not significantly affect alpha or beta microbiota diversity, it did induce different changes compared to the control in the bacterial genera composition of the colonic microbiota depending on the degree of almond processing: (1) whole raw almonds increased *Dialister* (*p* = 0.007), (2) roasted whole almonds increased *Lachnospira* (*p* = 0.03), and (3) roasted, chopped almond consumption increased the relative abundances of *Lachnospira, Roseburia*, and *Dialister* (*p* ≤ 0.05). Generally, these almond-related microbiota microflora are commensal bacteria known to produce short-chain fatty acids, especially butyrate, which can have positive effects on colonic health, immunity maintenance, and anti-inflammatory properties, but the health effects are not completely understood. Lastly, there were no differences in microbiota composition between almond butter and the almond-free control. 

Liu et al. [97], in a parallel RCT, randomized 48 young adults (24 males and 24 women, age 18–24 years, mean BMI 20) into an average 2400 kcal diet enriched with: 56 g of whole roasted almonds, 10 g of almond skins, or 8 g of fructooligosaccharides (control), divided into two daily servings for 6 weeks. Whole roasted almonds and almond skins significantly increased fecal populations of symbiotic *Bifidobacterium* and *Lactobacillus* and decreased populations of the pathogen *Clostridium perfringens.* That was directionally similar to fructooligosaccharides, but there were no significant changes in *E. coli* in any diet (Figure 39). After 6 weeks, almonds, almond skins, and fructooligosaccharides induced significantly reduced fecal nitro-reductases and beta-glucuronidase activity and increased fecal beta galactosidase activity, which are indicators of a healthy colonic microbiota. All treatments maintained fecal pH at a healthy level of <6.5 throughout the trial. This study confirmed the prebiotic activity of almonds in younger adults. 

Dhillon et al. [98], in a parallel RCT, randomly assigned 73 predominately breakfast-skipping college freshmen (41 women and 32 men, age 18 years, mean BMI 25) to a diet isocalorically enriched with whole roasted almonds (57 g/day) or isocaloric Graham cracker snacks for 8 weeks. Almond snacking increased quantitative alpha-diversity Shannon index by 3% and Chao1 index by 8% compared to the cracker group after 8 weeks (*p <* 0.05). Additionally, almond snacking decreased the overall abundance of the pathogenic bacterium *Bacteroides fragilis* by 48% (*p <* 0.05). In the same cohort of subjects, glucose tolerance and postprandial insulin sensitivity were enhanced with the almond snack, suggesting that improved colonic microbiota and carbohydrate metabolism may be inter-related. Overall, this trial indicated that the consumption of almonds improved the diversity and composition of the colonic microbiota that appear to be linked to improved metabolic health markers. 

Ren et al. [59], in a parallel RCT, randomly assigned 45 subjects with T2 diabetes (44% men, mean age 72 years, mean BMI 23) to one of two isocaloric diets: (1) an almond-based low-carbohydrate diet (56 g of almonds replacing 150 g of carbohydrate foods) or (2) a control low-fat diet (25% energy from fat) diet for 12 weeks with dietary counseling. Both diets significantly increased alpha diversity for Chao and PD index of the colonic microbiota compared to baseline (*p* < 0.05), which could promote long-term function resilience to metabolic stressors. The almond-based low-carbohydrate diet significantly increased the relative abundance of SCFA-producing bacteria *Roseburia* (*p* < 0.05), *Ruminococcus* (*p <* 0.05), and *Eubacterium* (*p* < 0.01).

Burns et al. [99], in a crossover RCT, randomized 29 parents (83% women, mean age 35 years) and their 29 children (48% girls, mean age 4 years) to one of two diets: (1) habitual diet with either 42.5 g of roasted almonds or almond butter for parents or 14 g of roasted almonds or almond butter for children, or (2) a control habitual diet for 3 weeks with a 6 week washout between the diet phases. The Healthy Eating Index scores in the almond diet were improved by 14% in both the parents and children compared to baseline. There were minimal changes in stool frequency or gastrointestinal symptoms with the almond-enriched diets. The colonic microbiota was relatively stable at the phylum and family levels, but genus changes happened with almond intake, especially in children. A stratified analysis found more changes from baseline in bacterial signatures in children consuming almonds compared with adults. No significant changes in immune biomarkers, including LPS-stimulated cytokines, salivary IgA, or serum antioxidants, were observed between the almond and control groups. A higher level of almonds for longer duration may be required to see immune effects. 

Darvishmoghadam et al. [100], in a double-blind RCT, randomized 50 diarrhea-prominent irritable bowel syndrome (IBS-D) patients (58% women, mean age 30 years) into an almond group (40 g of almond powder) or a placebo group (40 g of wheat and rice flour) for 20 days. Patients were assessed for bowel habit, pain severity, and frequency and bloating. Almonds did not improve any of the primary IBS outcomes. The diarrhea frequency and severity of pain were significantly increased in the almond powder group compared to the placebo group and baseline (*p* < 0.05). Despite the almond inclusion as a low FODMAP (fermentable oligo-, di-, mono-saccharides and polyols) food for potential IBS symptom relief, the intake of ground almond powder allowed for the release of oligo-fructans from the ground almonds, which are not released from whole or chopped almonds [5,6,7,102].

## 4. Discussion and Conclusions

### 4.1. Overview

This narrative review of 64 RCTs and 14 meta-analyses and/or systematic reviews presents a more in-depth analysis of almond clinical trials and their effects on weight measures, metabolic health biomarkers and outcomes, and colonic microbiota health than typically described in systematic reviews and/or meta-analyses. Almonds have one of the largest portfolios of RCTs on weight measures, metabolic health, and colonic microbiota of any food. These RCTs consistently support an important role for almonds in reducing body and fat mass, other weight measures, and promoting metabolic health as a premier snack for precision nutrition diets [103]. 

Snacking occurs in 93% of the USA population and contributes 23% of the total energy consumed in the day over an average of 2–3 snacking occasions/day. The most common snacks (e.g., cookies, candy, cakes, ice cream, frozen dairy, or chips) tend to have a relatively low healthy eating index (HEI) score of ≤59 (out of 100) but replacing these typical snacks in the American diet with almonds significantly increases the HEI score to 70 (*p* < 0.001) [104,105,106]. However, only about 6% of the American population are frequent tree nut consumers, with a usual intake of 44 g/day primarily as snacks, with almonds comprising about 40% of the tree nuts consumed [104,105]. Poor diet and snacking quality are a major determinant of the risk of overweight, obesity, and poor health outcomes [106]. For example, in the US, among children ages 2 to 19 years, 41% are overweight and obese, with the rate of obesity progressively increasing to 71% among adults aged 20 years and older [106]. When almonds are consumed as snacks or with meals, they help to promote higher diet quality than non-almond consumers experience by providing higher intakes of protein, healthier unsaturated fatty acids, fiber, folate, vitamin E, and magnesium, and lower starch, sugar, and sodium, in part because they tend to displace other lower dietary quality foods to improve weight control [3]. 

Poor quality diets are associated with progressively increasing weight gain, which can often lead to increased risk of central obesity and metabolic syndrome that can trigger a cascade of chronic disease risk factors, such as dyslipidemia, elevated blood glucose, hypertension, insulin resistance, increased systemic inflammation, and colonic microbiota dysfunction [106]. Obesity can also exacerbate severe adverse outcomes of influenza or COVID-19, associated with: (1) excessive abdominal fat, which pushes up on the diaphragm, causing the large muscle laying below the chest to restrict lung expansion and airflow, interfering with the delivery of oxygenated blood to the lower lung lobes, (2) increased risk for blood clotting, (3) weakened immune function because fat cells infiltrate immune organs such as the spleen, bone marrow, and thymus, making them less effective, and (4) elevated chronic, systemic low-grade inflammation, as fat cells are a major source of inflammation cytokines [107,108,109]. 

### 4.2. Almonds and Body Mass and Composition 

Although nuts are higher in ED with the potential to increase the risk of unintended body weight gain, RCTs consistently show that increased adiposity is not a risk with the daily intake of nuts. Recent systematic reviews and meta-analyses of nut RCTs showed that almonds are the only nut that slightly but significantly lowers mean body and fat mass compared to control nut-free diets to help reduce the risk of being overweight or obese, even when multiple daily servings are consumed [24,25,26,27,28]. Additionally, almonds are also among the best nuts to improve BMI, WC, abdominal, truncal, and visceral fat [24,25,26,27,28,42,43,53,56,62,65]. Almond body mass and composition management mechanisms include: suppression of hunger and desire to eat sensations, compensation for added energy from almonds by reducing energy intake of other foods, increased fecal excretion of macronutrients, especially fat, leading to a lower net ED compared to the Atwater estimates, inhibition of de novo lipogenesis associated with better glycemic control, and higher resting energy expenditure [5,6,7,29,30,31,32,33,34,35,36,37,38,39,40,102]. In addition, Cassady et al. [110] found that the number of times almonds are chewed can affect the body mass control mechanism, with 25 to 40 chews increasing fullness compared to 10 chews, and 10 chews increasing fecal energy excretion more than after 40 chews. 

Almond-based LCDs with energy deficits of 250 to 500 kcals/day generally have similar body mass loss than nut-free control LCDs [62,63,64]. However, LCDs with 1000 kcal deficits showed that almond-based LCDs significantly reduced BMI, body and fat mass, waist, and hip and/or waist–hip ratio compared to the control LCDs [65,66,67], which appears in part due to the fact that highly energy-restricted diets, such as 1000 kcal deficits, can promote suboptimal loss of body mass associated with increased colonic microbiota dysbiosis as a result of low prebiotic fiber intake, which can adversely alter production of gut peptides and SCFAs associated with satiety, food energy intake, and lipid metabolism, leading to a slower rate of weight loss, which can be corrected with prebiotic fiber enrichment such as that provided by almonds [1,2,10,92,93,94,95,96,97,98,101,111]. 

### 4.3. Almonds and Metabolic Health Biomarkers and Outcomes

Almonds are rich in healthy unsaturated fats, fiber, proteins, vitamin E and B-vitamins, calcium, magnesium and copper, and phytosterols and polyphenols, and low in available carbohydrates, which can support various healthy biomarkers associated with reduced chronic disease risk [2,112,113,114,115]. RCTs and systematic reviews and meta-analyses of RCTs are building the evidence that almonds are a healthy adjunctive dietary option for the prevention and management of CVD and T2 diabetes, which are leading global causes of pre-mature death and medical costs. Almonds consistently improve blood lipid profiles, including significant lowering TC, LDL-C, non-HDL-C, VLDL-C, Apo-B, and small LDL-C particles, and modest reductions in diastolic BP, which collectively translates into significant and clinically important reductions in the atherogenic index. However, almonds have inconsistent and/or insignificant beneficial effects on endothelial function, inflammatory markers, glycemic control, HbA1c, flow-mediated dilation, systolic BP, mean arterial pressure, or pulse pressure, which need to be studies in larger and longer-term trials. Several emerging RCTs suggest that almonds may play a role in improving working and visual memory and help to reduce facial wrinkle severity and width. Additionally, emerging data indicates that the intake of nuts such as almonds is associated with improved health status of older adults (e.g., longer telomere length and lower risk of sarcopenia) [116]. 

Two of the most important nutritional attributes of almonds associated with the reduced risk and better management of CVD and T2 diabetes are the healthy unsaturated lipids and low glycemic index (GI) and glycemic load (GL). The US Food and Drug Administration approved a qualified health claim that eating 1.5 ounces (42.5 g)/day of tree nuts such as almonds as part of a diet low in saturated fat and cholesterol may reduce the risk of CHD [117]. A medical cost savings and quality of life analysis study in the US population using current US population CVD rates and the probabilities of elevated LDL-C increasing risk, disability, and death from CVD, found that increasing daily consumption of almonds up to 42.5 g/day could significantly reduce annual US CVD costs by about 25% and increase annual improvement in quality adjusted life years by 2% compared to <3 g almond/day average intake [118]. Other cost-benefit analyses show with high confidence that increased intake of foods such as almonds that lower glycemic index and glycemic load are causally correlated with reduced incidence of and medical costs related to T2 diabetes, which can reach 10% of the national health expenditures [119,120]. It is estimated that 20% of these T2 diabetes healthcare costs are related to the addition to the diet of high glycemic index (GI) and glycemic load (GL) foods, which are reduced with almond intake, an essentially zero GI and GL food [1]. Although Becerra-Tomas et al. [121], in a systematic review and meta-analysis of 8 observational studies, found no association between total nuts, tree nuts, or peanuts and risk of T2 diabetes. Jenkins et al. [112], in an RCT, showed that nuts as a replacement for carbohydrates improves glycemic control, including lowering HbA1c, and reduces CVD risk by lowering the number of small LDL particles and levels of TC, LDL-C, non-HDL-C, and Apo-B in individuals with T2 diabetes. 

Almonds are a good source of polyphenols and α-tocopherols which could reduce oxidative stress to potentially support primary and secondary CVD prevention, improving cognitive performance and reducing Alzheimer’s disease (AD) risk [2,122,123]. The antioxidant activity of almonds has been best evaluated in two RCTs on habitual smokers, which showed some improvements of antioxidant biomarkers due to polyphenol compounds in the almond skins and suggested some antioxidant protection in smokers [124,125]. A meta-analysis of 31 articles indicated that individuals with AD, age-related cognitive impairment, or mild cognitive decline had lower circulatory concentrations of α-tocopherol compared to healthy controls [126]. Additionally, an analysis of 115 deceased and autopsied cases in the Rush Memory and Aging Project showed that higher levels of brain tocopherols such as α-tocopherol were related to lower activated microglia density in the cortical brain, which might protect against AD to lower levels of inflammation, amyloid, and neurofibrillary tangle severity [127]. 

### 4.4. Almonds and Colonic Microbiota 

There is a growing body of RCTs that are emerging to support almond’s role in promoting a healthy microbiota [59,94,95,96,97,98,99,100]. The colonic microbiota appears to play a major role in human metabolic health, and it is primarily controlled by the nutritional quality of the diet [128,129]. Colonic microbiota can be modulated positively or negatively by different lifestyle and dietary factors and impact the risk of developing obesity, chronic diseases (e.g., diabetes, CVD, and metabolic syndrome features), and infectious diseases. Moreover, microbial metabolites can induce epigenetic modifications (i.e., changes in DNA methylation and micro-RNA expression), with potential implications for health status and susceptibility to obesity. Microbial products, such as SCFAs, may affect metabolism by regulating appetite, lipogenesis, gluconeogenesis, inflammation, and other functions. Intestinal dysbiosis can change the functioning of the intestinal barrier and the gut-associated lymphoid tissues (GALT) by allowing the passage of structural components of bacteria, such as lipopolysaccharides (LPS), which activate inflammatory pathways that may contribute to the development of CVD, insulin resistance, and alter the production of gastrointestinal peptides related to satiety, food intake, and body fat metabolism. In addition, in CHD patients, a healthy colonic microbiota was shown to have anti-hyperuricemia effects by increasing uric acid excretion and regulating uric acid absorption by certain transporters, and the intake of 10 g/day of almonds soaked overnight and consumed before breakfast was shown to reduce serum uric acid by 18% over 12 weeks [130,131]. 

### 4.5. Strengths and Limitations

This comprehensive narrative review has many strengths, including: (1) being the first almond review that provides in-depth summaries of all the almond RCTs on body mass and composition, metabolic health biomarkers and outcomes, and colonic microbiota in one publication, (2) organization of the almond RCTs into subgroups to help reveal new insights or data trends, (3) extensive use of figures and tables to highlight important trial results, and (4) an overview of all almond RCTs to help chart a better plan for more future high-quality RCTs. This review also has some limitations, including: (1) most of the almond RCTs have a relatively small number of subjects and short duration, which may lead to excessive heterogeneity and mask some other metabolic outcomes in non-blood lipid biomarkers, and (2) this narrative review did not score the quality of each study or provide a quantitative, scientific synthesis of all RCTs into a statistical overall mean or analysis of subgroups, effect sizes, and heterogeneity, independently of what was provided by the specific RCTs reviewed.

### 4.6. Potential Future RCTs

The Almond Board of California has already begun to address these limitations with an RCT in-progress, including funding a 134 subjects for 9 months studying body mass and composition, and metabolic health during 30% energy-restricted diets [80]. However, there is also a need to support similar quality RCTs in non-energy-restricted RCTs. Additionally, since there is only one small RCT which evaluated young children and one RCT in college freshmen, it would be important to perform more studies on the effects of almonds on body mass and composition, and metabolic health biomarkers in toddlers through teenagers, because being overweight or obese and having poor metabolic health are growing health concerns in this age group. Additionally, it would be important to build on the facial wrinkle reduction research and expand research into other effects of almonds on healthy aging biomarkers.

## Figures and Tables

**Figure 1 nutrients-13-01968-f001:**
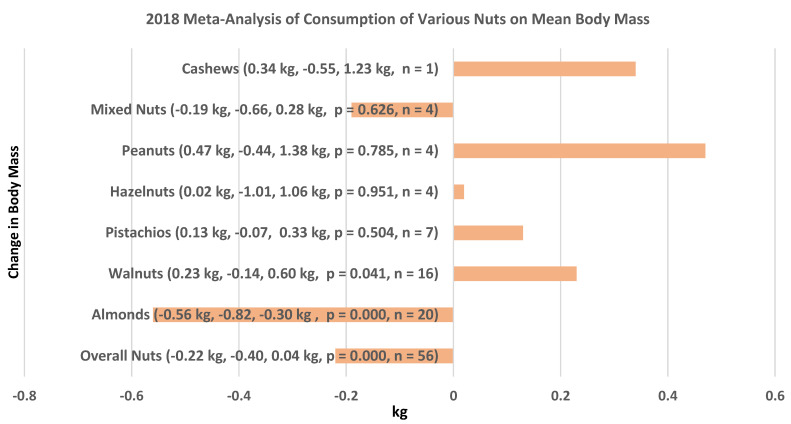
The effects of specific nut-enriched diets and overall nut intake on change in body mass compared to control diets based on a meta-analysis of 62 RCTs [24].

**Figure 2 nutrients-13-01968-f002:**
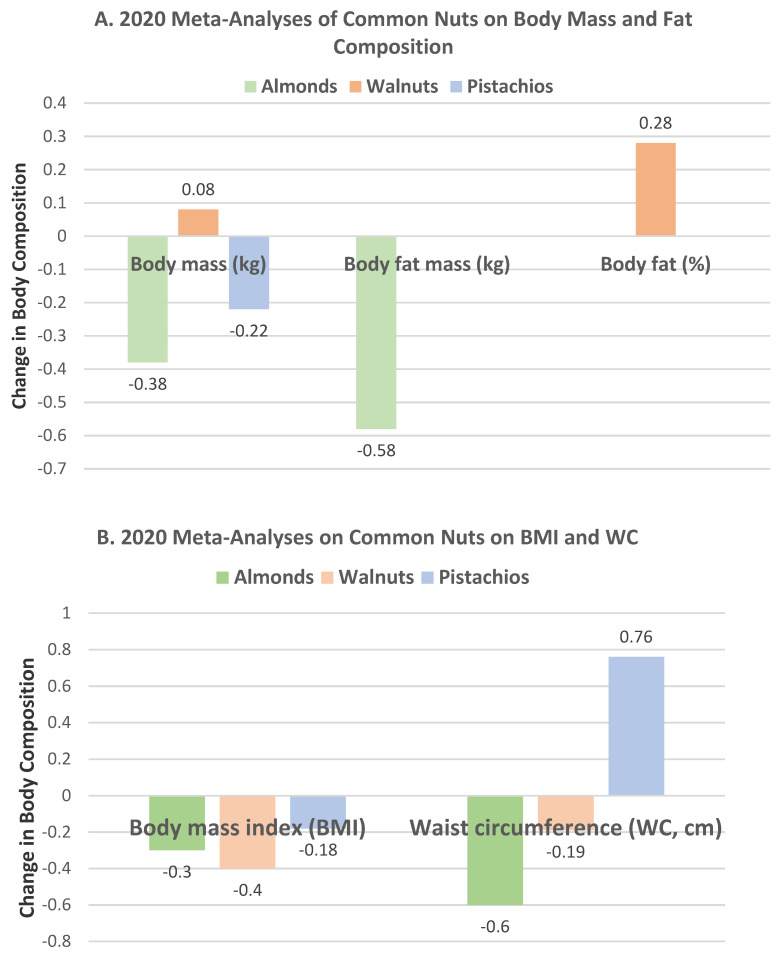
(**A**) Effect of nut-enriched diets compared to control diets on mean changes in body and fat mass (almonds body mass *p* = 0.007 and fat mass *p* ≤ 0.001; walnuts body mass *p* = 0.193 and body fat *p* = 0.476; pistachios body mass *p* = 0.141) [25,26,27]. (**B**) Effect of nut-enriched diets compared to control diets on mean changes in BMI (almonds *p* = 0.10, walnuts *p* = 0.164, and pistachios *p* < 0.001) and waist circumference (WC) (almonds *p* = 0.078, walnuts *p* = 0.651, and pistachios *p* = 0.087) [25,26,27].

**Figure 3 nutrients-13-01968-f003:**
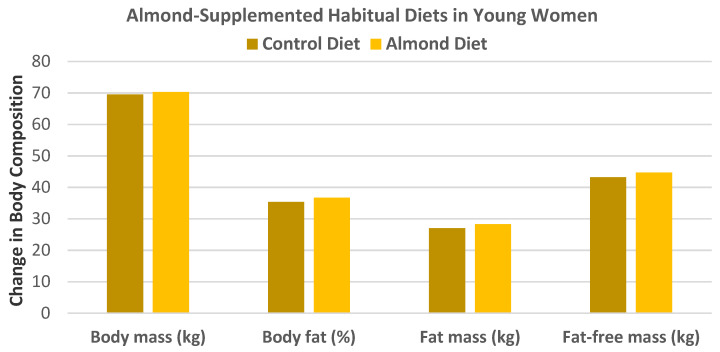
Changes in body weight and composition measures after 60 g (344 kcals) raw almonds/day are added to the habitual diets of young women compared to the habitual diet control over 10 weeks (*p* > 0.05) [31].

**Figure 4 nutrients-13-01968-f004:**
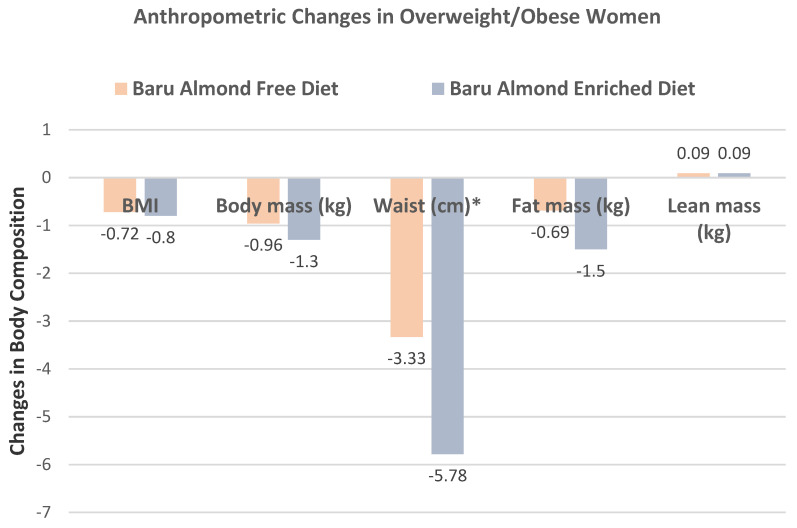
Change in body weight measures from baseline to 8 weeks for baru almond-enriched diets and control diets in overweight/obese women (* *p* = 0.03) [35].

**Figure 5 nutrients-13-01968-f005:**
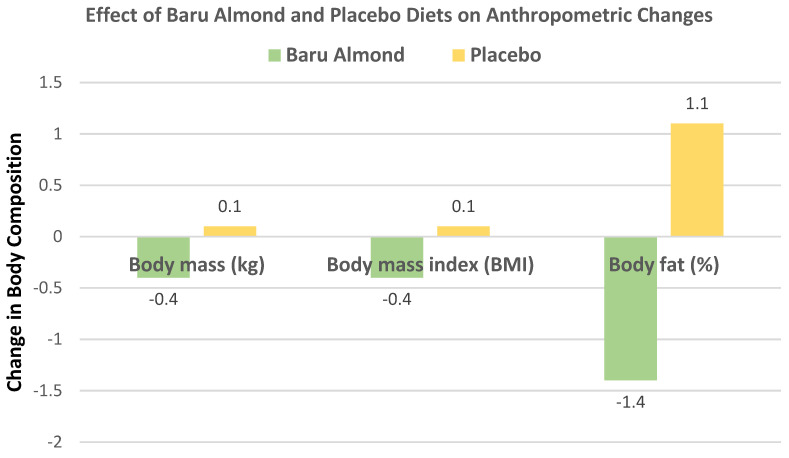
Effect of baru almonds and placebo on body components from baseline to 6 weeks (body mass (*p* = 0.07), BMI (*p* = 0.06), and body fat (%, *p* = 0.10) [36].

**Figure 6 nutrients-13-01968-f006:**
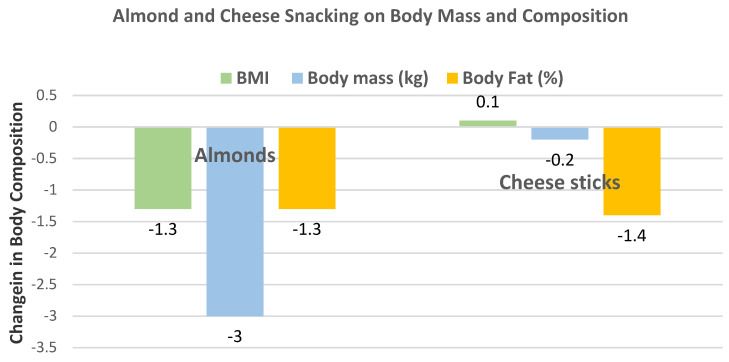
Effects of 5 servings per week of almonds or cheese sticks on body mass and composition parameters from baseline to 12 weeks in obese individuals with T2 diabetes (almonds BMI *p* = 0.047, body mass *p* = 0.083, and body fat *p* = 0.853) compared to cheese sticks [39].

**Figure 7 nutrients-13-01968-f007:**
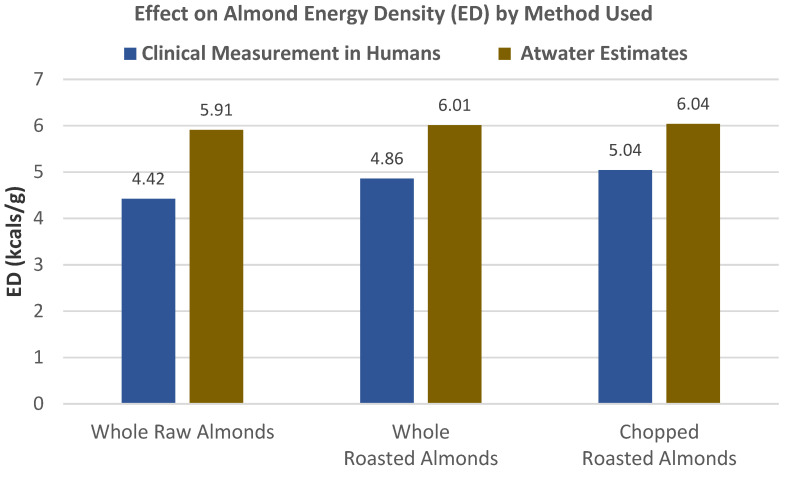
Differences in ED between Atwater estimates and direct clinical measures of three forms of almonds when added to the typical American diet [6].

**Figure 8 nutrients-13-01968-f008:**
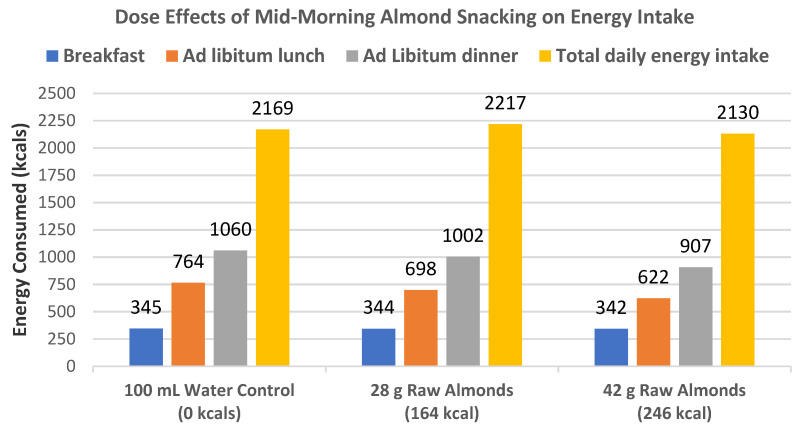
Dose–response effects of mid-morning raw almond snacking on subsequent meal energy intake, leading to an insignificant change in daily total energy intake compared to the water control in healthy women [7].

**Figure 9 nutrients-13-01968-f009:**
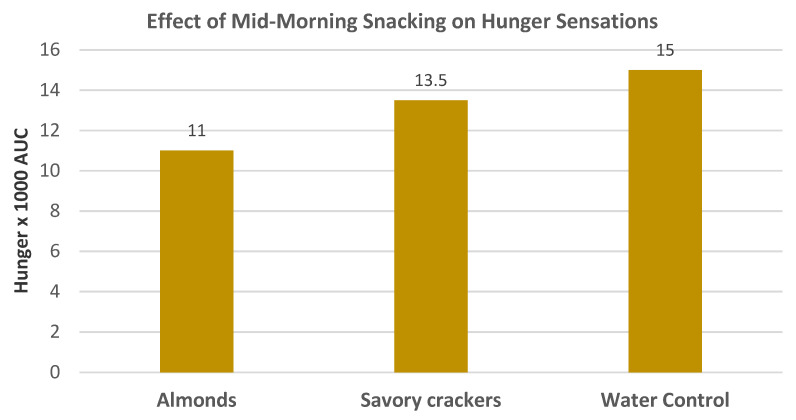
Effect of mid-morning snack on subjective ratings of hunger through dinner for almonds, savory crackers, and water control (almond vs. water *p* < 0.001 and almond vs. cracker *p* < 0.05) [40].

**Figure 10 nutrients-13-01968-f010:**
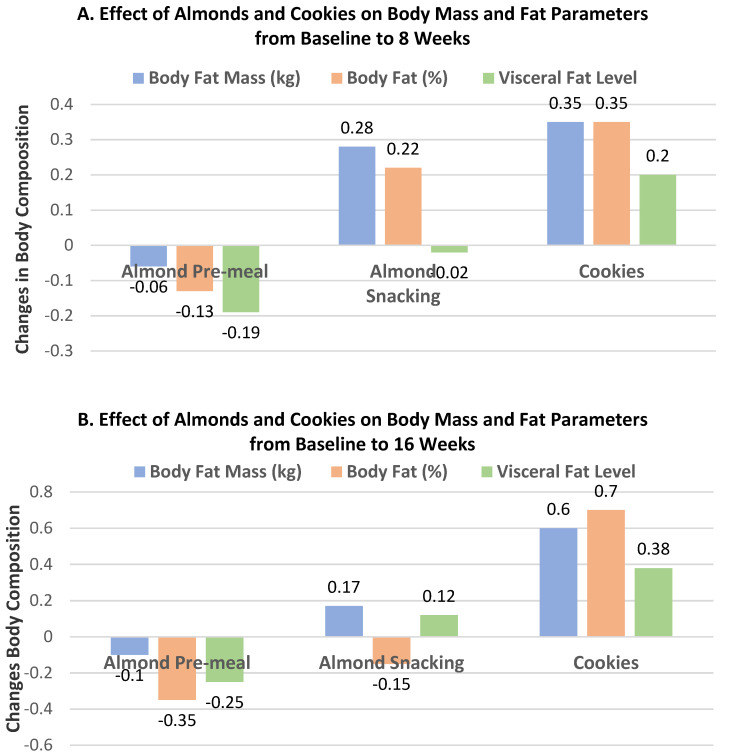
(**A**,**B**) Changes in body fat composition from baseline to 8 and 16 weeks for almond pre-meal snacking, between meal snacking and cookie snack control in young healthy adults [42].

**Figure 11 nutrients-13-01968-f011:**
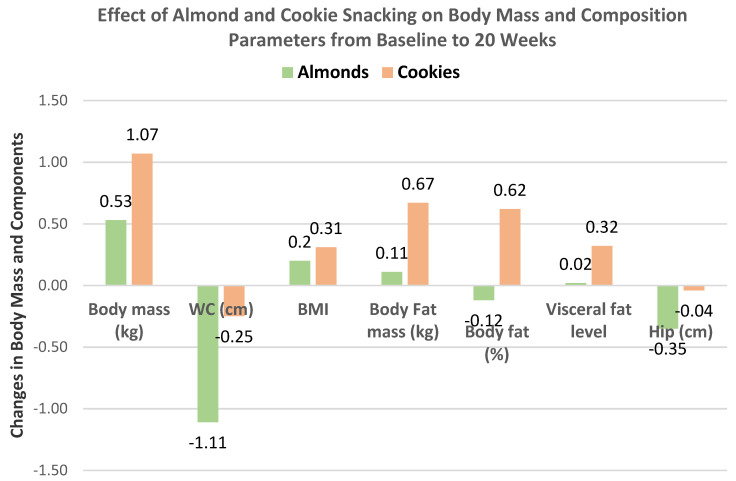
Changes in body mass measures for almonds and control cookie snacking from baseline to 20 weeks [43].

**Figure 12 nutrients-13-01968-f012:**
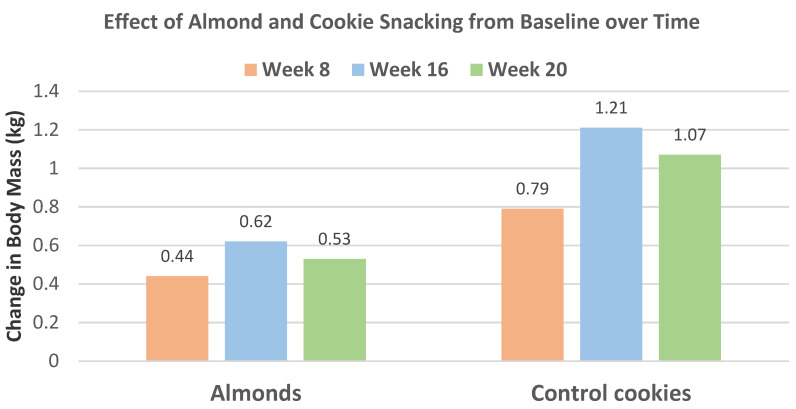
Changes in body mass for almonds and control cookies over time (almonds *p* = 0.162 and cookies *p* = 0.001) [43].

**Figure 13 nutrients-13-01968-f013:**
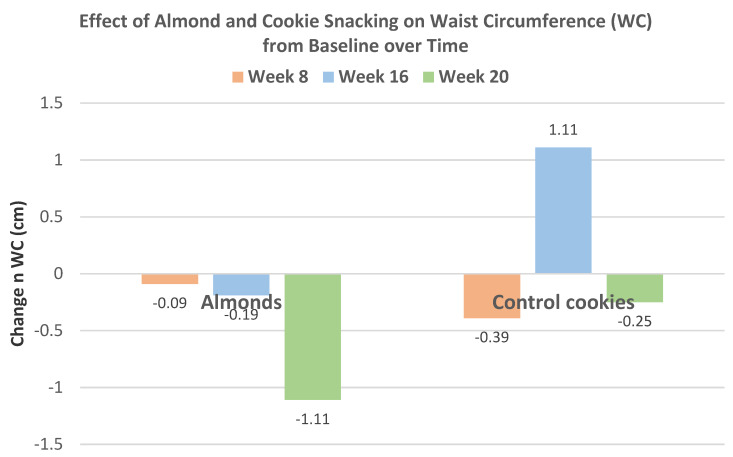
Changes in WC for almonds and control cookies over time (almonds *p* for time effects = 0.013 [43].

**Figure 14 nutrients-13-01968-f014:**
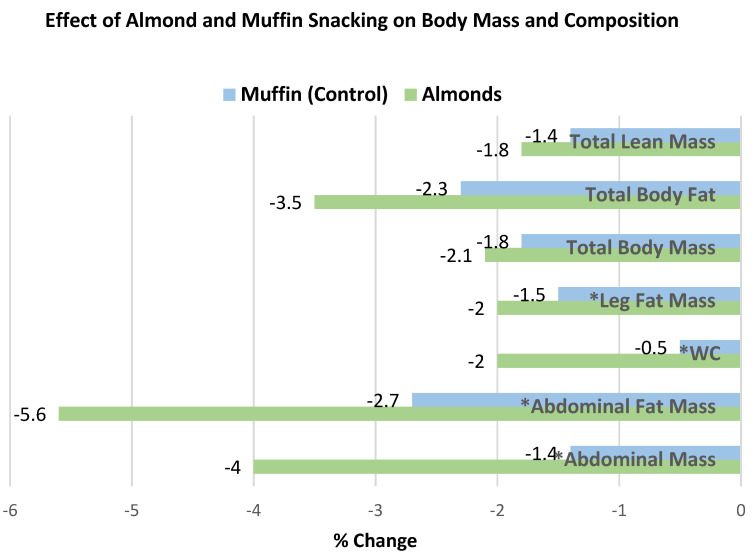
Changes in % body mass and composition outcomes for raw almonds (42.5 g /day) compared to muffin control in identical heart-healthy diets from baseline to 6 weeks (* *p* < 0.02) [53].

**Figure 15 nutrients-13-01968-f015:**
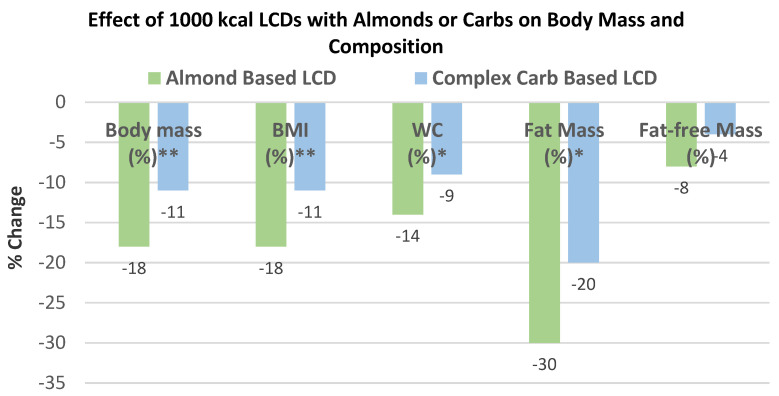
Effect of 1000 kcal almond- and carbohydrate-based LCDs on weight measures from baseline to 24 weeks (** *p* < 0.0001 and * *p* < 0.05) [65].

**Figure 16 nutrients-13-01968-f016:**
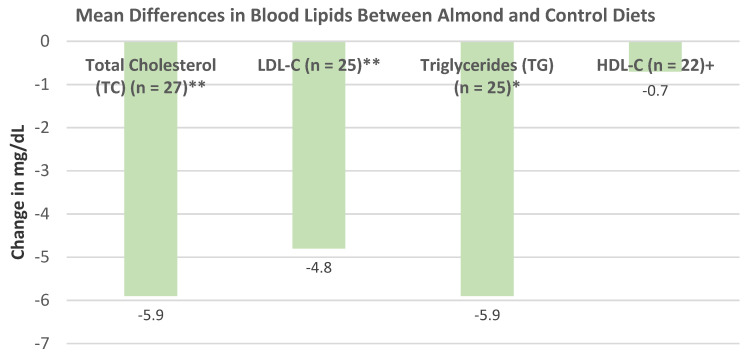
Meta-analysis on mean changes in blood lipids between almond and control interventions (** *p* ≤ 0.001, * *p* = 0.042, + *p* = 0.207) [69].

**Figure 17 nutrients-13-01968-f017:**
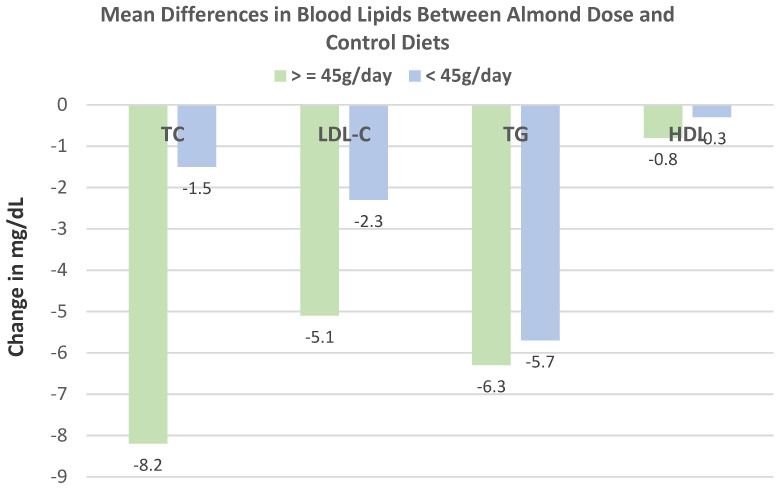
Meta-analysis on mean differences in blood lipids between dose of almonds and control diets for ≥45 g doses of TC and LDL-C *p* = 0.001, and TG and HDL-C *p* ≤ 0.188, and <45 g doses of TG *p* = 0.199 and TC, LDL-C and HDL-C *p* ≥ 0.470 [69].

**Figure 18 nutrients-13-01968-f018:**
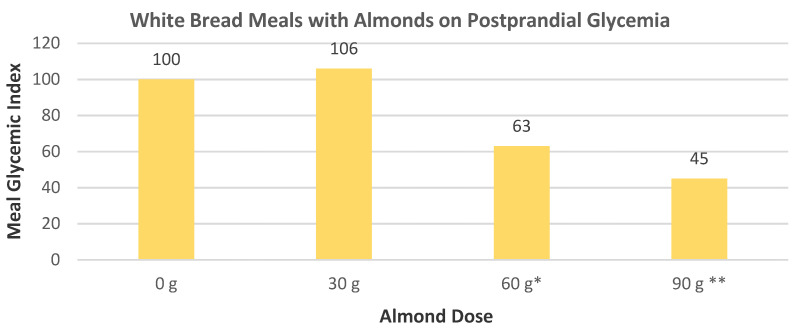
Dose–response effect of increasing level of almonds consumed with white bread on glycemic index over 120 min (* *p* < 0.02, ** *p* < 0.01) [74].

**Figure 19 nutrients-13-01968-f019:**
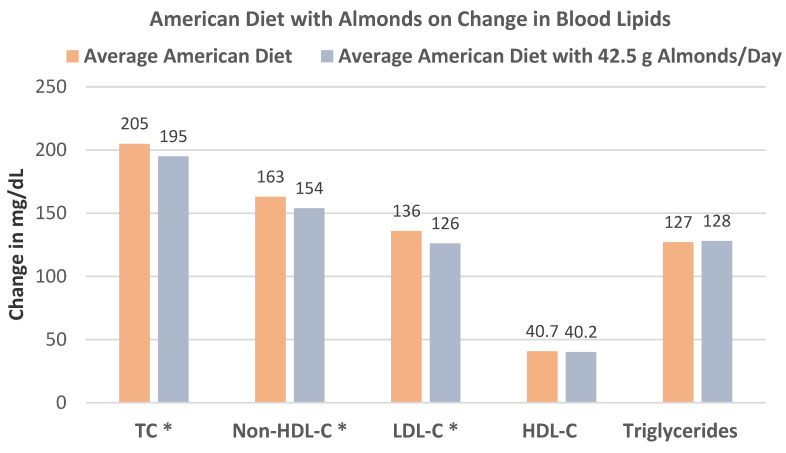
Effect of the average American diet and the average American diet with almonds on blood lipids in overweight and obese individuals from baseline to 4 weeks, with the inclusion of almonds lowering LDL-C by 4%, non-HDL-C by 5%, and LDL-C by 7% (* *p* <0.05) [77].

**Figure 20 nutrients-13-01968-f020:**
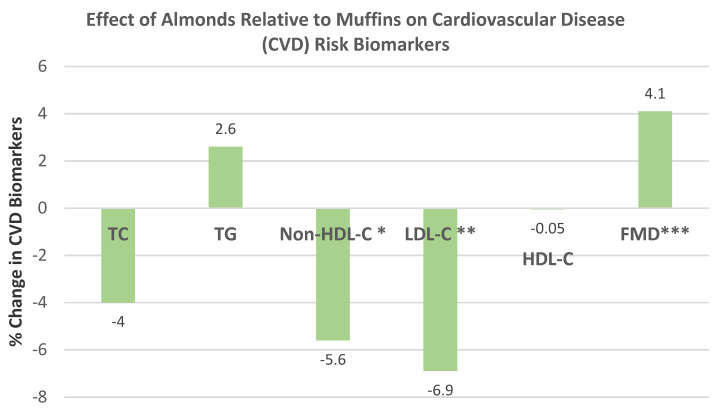
Changes in % mean CVD risk biomarkers for the almond diet compared to the muffin control diet from baseline to 6 weeks (* *p* = 0.037, ** *p* = 0.017, *** *p* < 0.00005) [45].

**Figure 21 nutrients-13-01968-f021:**
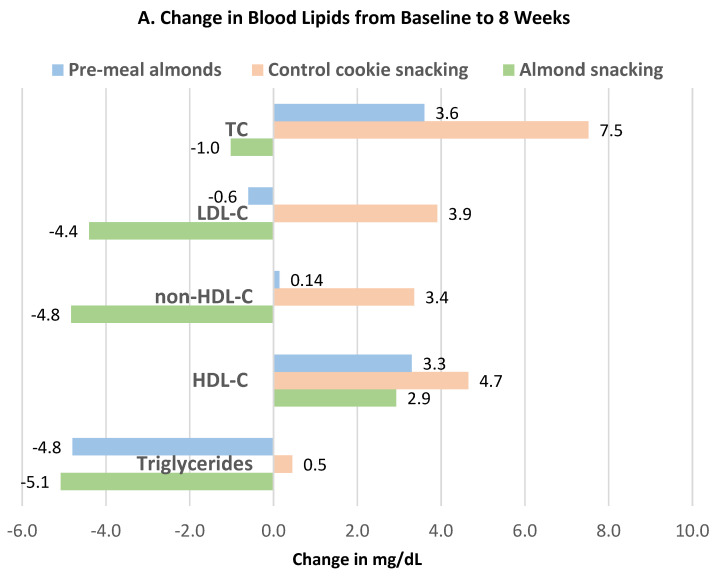
(**A**,**B**) Change in blood lipids from pre-meal almonds, almond snacks, and cookie snack control from baseline to 8 and 16 weeks [42].

**Figure 22 nutrients-13-01968-f022:**
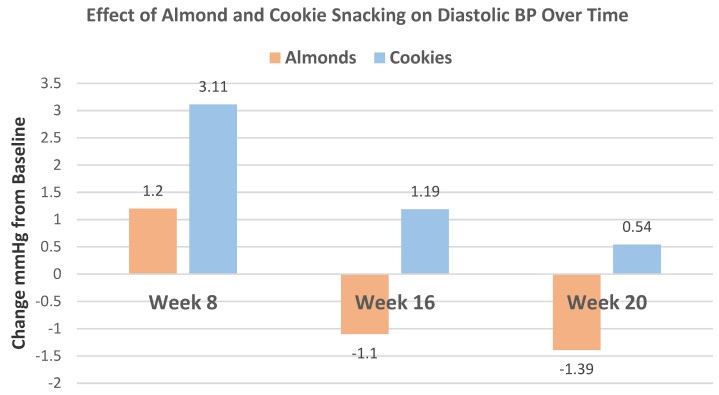
Effect of almond and cookie snacking on diastolic BP over time (almonds *p*-trend = 0.012 and control cookies *p*-trend = 0.135) [43].

**Figure 23 nutrients-13-01968-f023:**
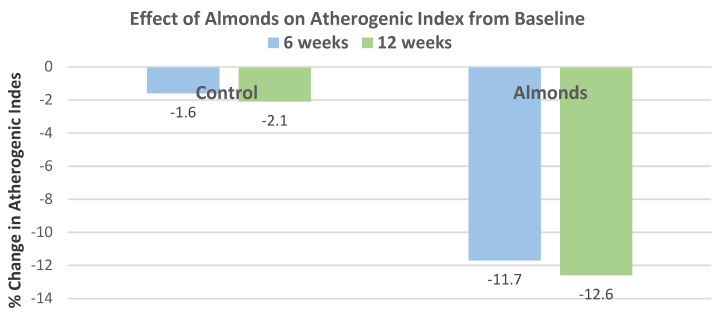
Effects of 10 g of almonds soaked overnight in water and consumed before breakfast on atherogenic index compared to no almond control (*p* = 0.05) [37].

**Figure 24 nutrients-13-01968-f024:**
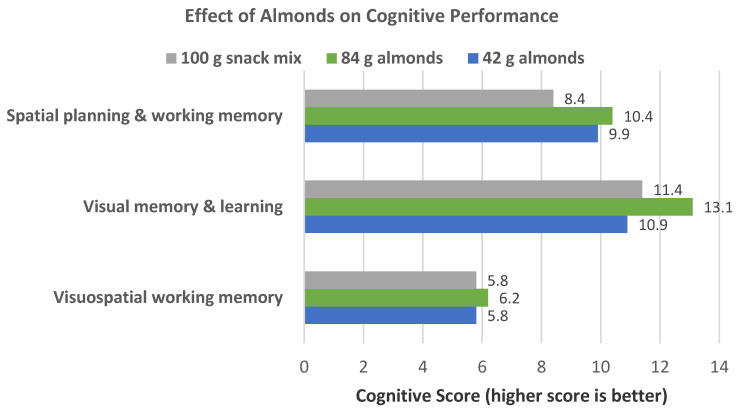
Change in cognitive measures after 6 months for almonds and mixed snacks with 84 g of almonds (3 servings/day) significantly improving working and visual memory cognitive measures (*p* < 0.017) compared to the control snack mix [81].

**Figure 25 nutrients-13-01968-f025:**
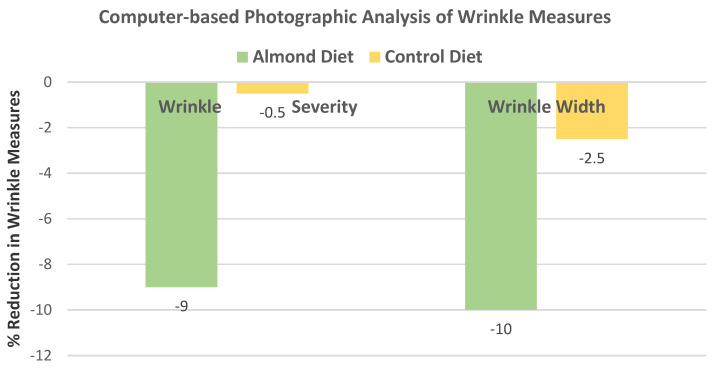
Almond group compared to carbohydrate-rich group on wrinkle measures (*p* < 0.02 for both almond wrinkle measures) from baseline to 16 weeks [82].

**Figure 26 nutrients-13-01968-f026:**
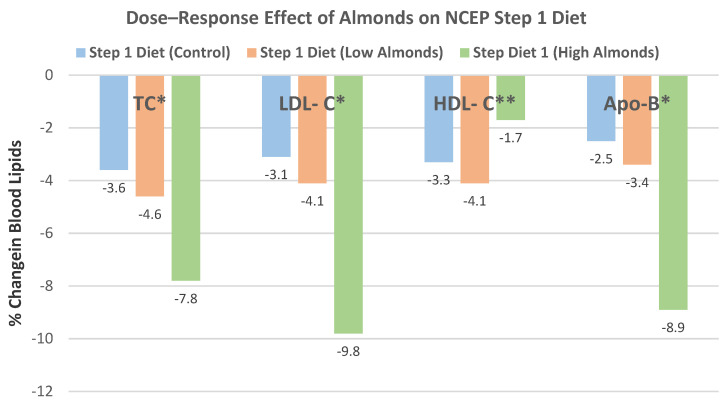
Dose–response effects on blood lipids when almonds are added to the NCEP Step 1 diet from baseline to 4 weeks: high almond intake significantly improved non-HDL-C lipids * (*p* < 0.001) and insignificantly changed HDL-C ** (*p* = 0.09) [50].

**Figure 27 nutrients-13-01968-f027:**
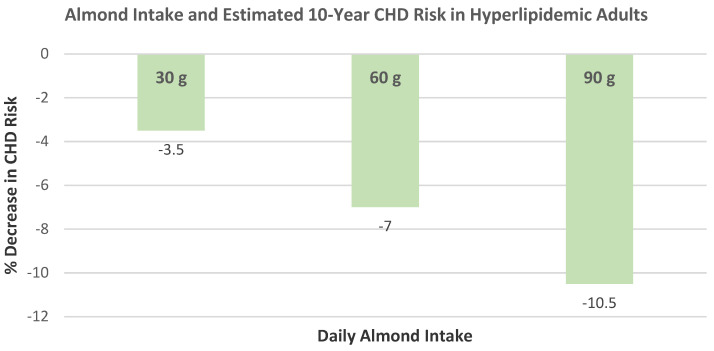
Regression analysis of correlation between almond intake and percentage decrease in the estimated 10-year Framingham coronary heart disease (CHD) risk score in hyperlipidemic adults [85].

**Figure 28 nutrients-13-01968-f028:**
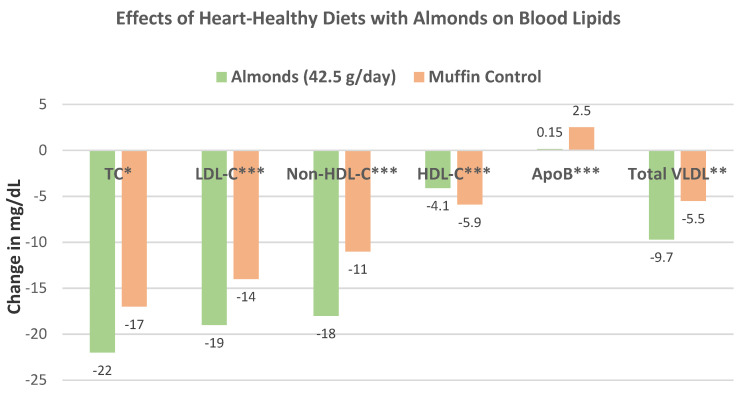
Effect of heart-healthy diets with raw almonds or a muffin control in hyperlipidemic individuals from baseline to 6 weeks, * *p* = 0.04, ** *p* = 0.02, and *** *p* < 0.01 [53].

**Figure 29 nutrients-13-01968-f029:**
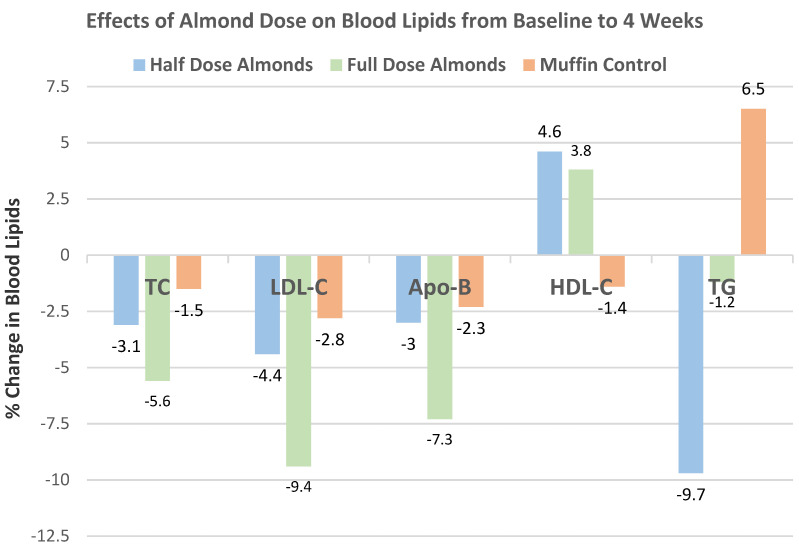
Dose–response effects of almond-enriched NCEP Step 2 diets from baseline to 4 weeks: (1) full-dose almonds significantly reduced TC, LDL-C, and Apo-B (*p* < 0.001), (2) half-dose almonds reduced TC, LDL-C, and Apo-B (*p* < 0.057), (3) both almond doses significantly increased HDL-C (*p* < 0.047), and (4) the muffin control only significantly increased triglycerides (*p* = 0.031), with no significant changes in non-HDL or HDL-C blood lipids [51].

**Figure 30 nutrients-13-01968-f030:**
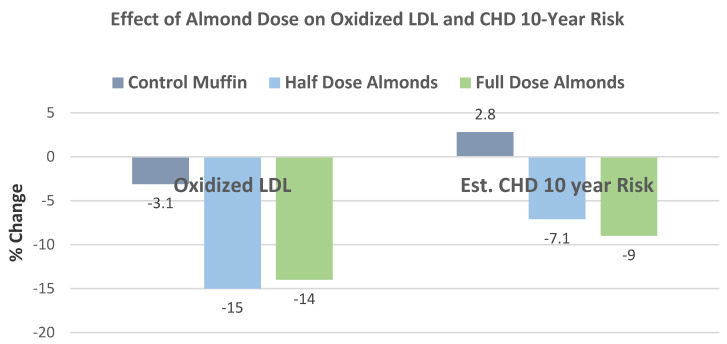
Dose–response effects of almond NCEP Step 2 diets from baseline to 4 weeks: (1) Both doses of almonds significantly reduced LDL oxidation compared to baseline and muffin control (*p* < 0.001). (2) The full almond dose significantly reduced estimated 10-year CHD risk (*p* ≤ 0.029) [51].

**Figure 31 nutrients-13-01968-f031:**
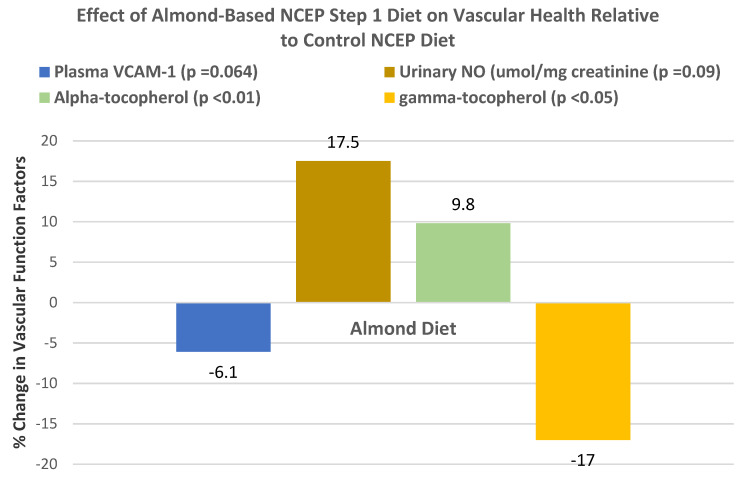
Effect of almond National Cholesterol Education Program (NCEP) Step 1 diets on vascular function factors and tocopherols relative to control NCEP Step 1 diet from baseline to 6 weeks [86].

**Figure 32 nutrients-13-01968-f032:**
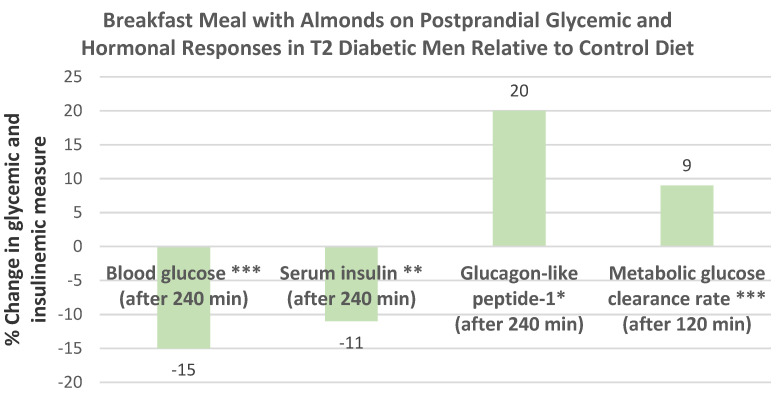
Postprandial % change in area under the curve (AUC) for blood glucose and insulin responses for isocaloric white bread plus almonds vs. white bread plus butter and cheddar cheese meals (* *p* = 0.283, ** *p* = 0.021, **** *p* ≤ 0.005) [90].

**Figure 33 nutrients-13-01968-f033:**
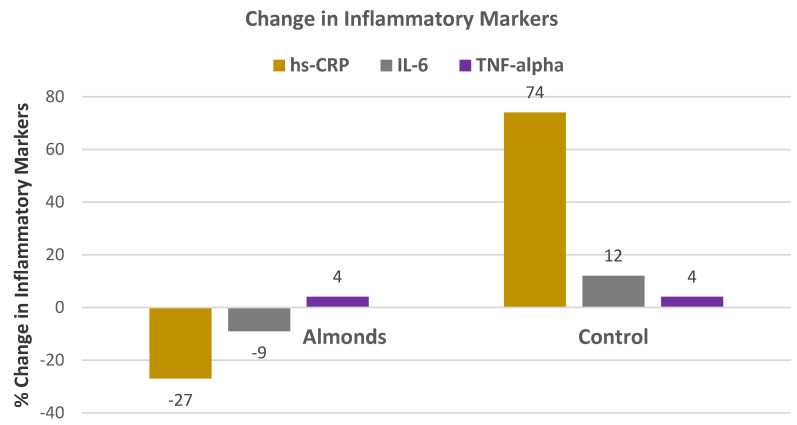
The almond snack significantly lowered hs-CRP (*p* = 0.029) and other inflammatory markers (*p* > 0.05) vs. the control diets from baseline to 6 weeks (*p* > 0.05) [34].

**Figure 34 nutrients-13-01968-f034:**
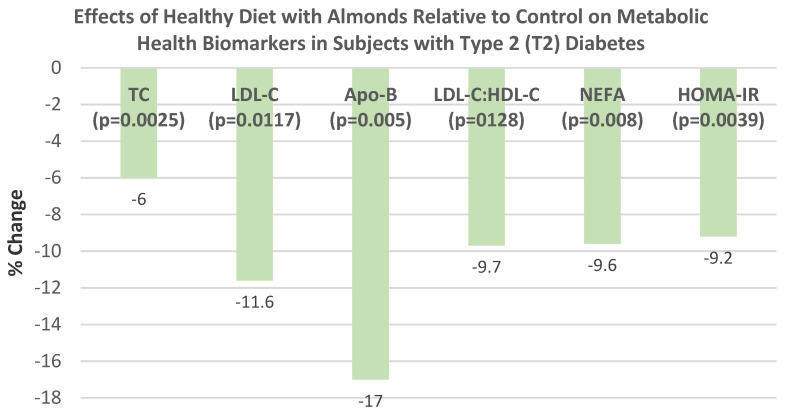
Effect of consuming an almond-supplemented NCEP Step 2 diet relative to a control in T2 diabetic subjects from baseline to 4 weeks [54].

**Figure 35 nutrients-13-01968-f035:**
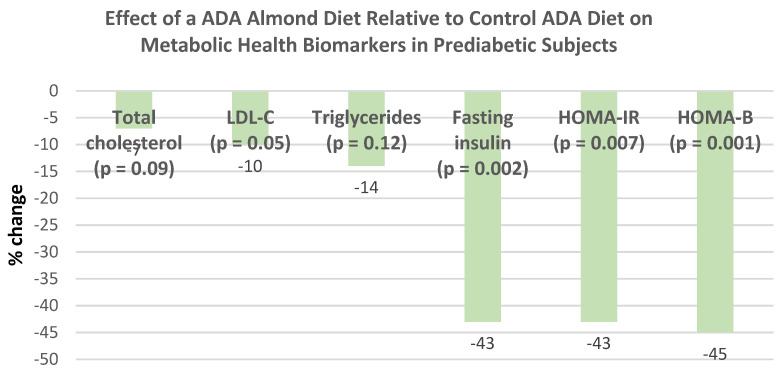
Change (%) in metabolic risk biomarkers between the almond diet and nut-free control diet for baseline to 16 weeks during a weight maintenance diet in prediabetic individuals [62].

**Figure 36 nutrients-13-01968-f036:**
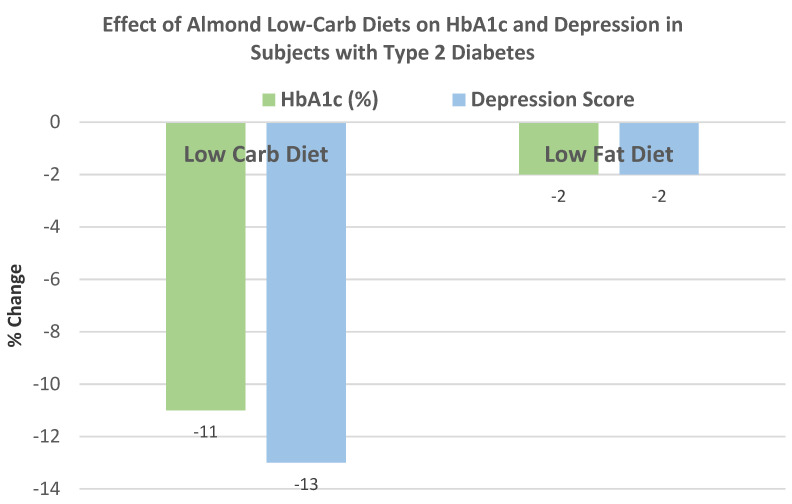
Comparison of HbA1c and the depression scores between almond low-carbohydrate and low-fat diets between baseline and 12 weeks (for almonds, *p* < 0.01 for both) [59].

**Figure 37 nutrients-13-01968-f037:**
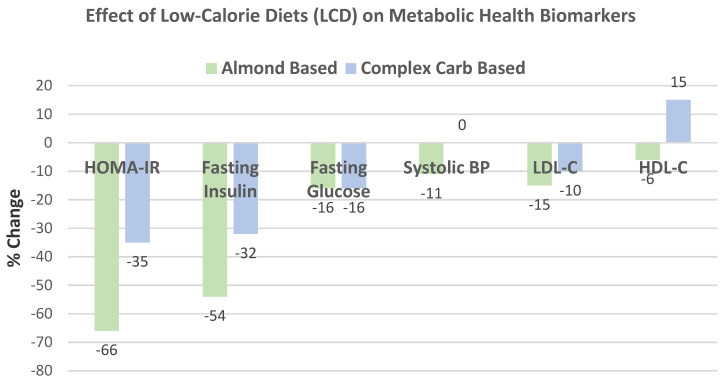
Summary of effects of almond-enriched and complex carb-based LCDs on metabolic outcomes from baseline to 24 weeks: almond vs. complex carb LCD change in HOMA-IR, fasting insulin and glucose, and LDL-C (*p* < 0.0001), systolic BP (*p* < 0.01), and HDL-C (*p* < 0.05) [65].

**Figure 38 nutrients-13-01968-f038:**
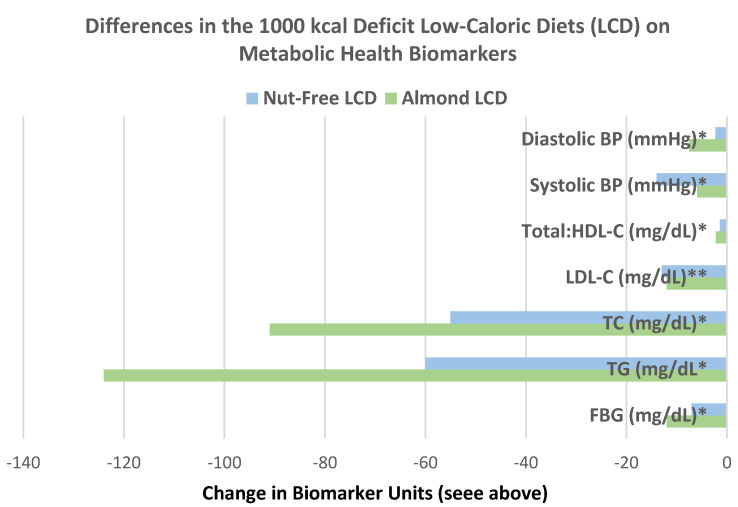
Difference in metabolic healthy biomarkers between almond and nut-free LCDs from baseline to 12 weeks (fasting blood glucose (FBG), triglycerides (TG), total cholesterol (TC), low-density lipoprotein cholesterol (LDL-C), and blood pressure (BP), **p* ≤ 0.001 and ** *p* = 0.002) [66].

**Figure 39 nutrients-13-01968-f039:**
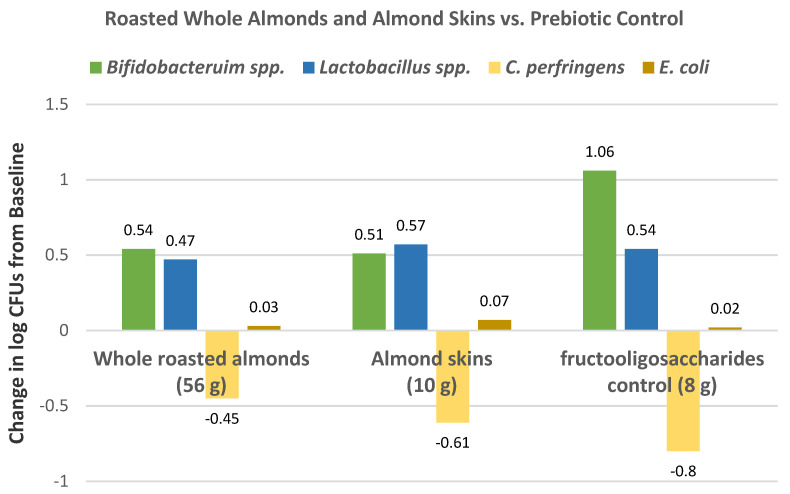
Change in colony-forming units (CFUs) from baseline to 6 weeks: (1) symbiotic bacteria *Bifidobacterium* and *Lactobacillus* increased (*p* ≤ 0.05 for all) and (2) pathogenic bacteria *Clostridium perfringens* decreased for fructooligosaccharides and almond skins (*p* < 0.05) and whole almonds (*p* < 0.1), and there were no significant changes in fecal *E. coli* (*p* > 0.05) [97].

**Table 1 nutrients-13-01968-t001:** Stratified overall mean change in anthropometric measures for nuts with and without dietary substitution guidance [23].

Outcome Variable	No Dietary Substitution	*p*-Value	Dietary Substitution	*p*-Value
Body mass (kg)	0.01 (−0.07, 0.08)	0.88	−0.01 (−0.11, 0.09)	0.82
Body mass index (kg/m^2^)	−0.01 (−0.08, 0.11)	0.80	0.00 (−0.12, 0.13)	0.95
Waist (cm)	0.01 (−0.09, 0.10)	0.85	0.01 (0.11, 0.13)	0.086
Body fat (%)	−0.05 (−0.19, 0.09)	0.45	−0.32 (−0.61, −0.03)	0.03

**Table 2 nutrients-13-01968-t002:** Stratified mean body mass (*n* = 11) and body mass index (*n* = 6) changes by almond dosage and trial duration (* *p* < 0.05) [28].

Outcomes	Overall	Subgroup Analysis of Dosage		Subgroup Analysis of Duration	
	All	≤42.5 g	>42.5 g	≤6 Weeks	>6 Weeks
Body mass (kg)	−1.39 (−2.5, −0.3) *	−0.17 (−2.35, 2.01)	−1.80 (−3.07, −0.54) *	0.10 (−2.12, 1.92)	−1.93 (−3.23, −0.63) *
Body mass index (kg/m^2^)	−0.33 (−1.1, 0.4)	0.22 (−2.18, 2.62)	−0.25 (−1.25, 0.74)	0.13 (−1.15, 1.40)	−0.33 (−1.51, 0.86)

**Table 3 nutrients-13-01968-t003:** Anthropometric changes from baseline to 8 weeks by snack group in older obese adults at risk for or with type 2 diabetes [47].

Outcome	Almond Snacking	Biscuit Snacking	*p*-Value
Body mass (kg)	0.78 ± 1.82	0.68 ± 1.33	0.88
Body fat mass (kg)	−0.14 ±1.61	0.39 ±1.34	0.77
Fat-free mass (kg)	0.91 ±1.73	0.40 ±1.25	1.00
Waist circumference (cm)	−0.25 ± 2.66	0.22 ± 2.44	0.91
Visceral adipose tissue (cm^2^)	−3.18 ± 15.9	−0.99 ± 19.4	0.97
Liver fat (%)	1.53 ± 4.64	1.88 ± 3.51	0.84
Subcutaneous adipose tissue (cm^2^)	0.46 ± 16.1	4.04 ± 16.1	0.78

**Table 4 nutrients-13-01968-t004:** Mean changes in weight measures for 500 kcal deficit almond-enriched and nut-free low-calorie diets (LCDs) [63].

Change in Measure	Almond	Nut-Free	*p*-Value
Body mass (kg)			
6 months	−5.5 ± 0.6	−7.4 ± 0.7	0.04
18 months	−3.7 ± 1.0	−5.9 + 1.0	0.12
Fat mass (kg)			
6 months	−3.7 ± 0.5	−5.0 ± 0.5	0.06
18 months	−3.0 ± 0.8	−4.0 ± 0.8	0.39
Lean mass (kg)			
6 months	−1.8 ± 0.3	−2.5 ± 0.3	0.22
18 months	−1.4 ± 0.4	−2.4 ± 0.4	0.09

**Table 5 nutrients-13-01968-t005:** Changes in blood lipids from baseline: National Cholesterol Education Program (NCEP) Step 3 diet with almonds vs. without almonds in subjects with statin therapy [58].

Outcome	NCEP Step 3 Diet Almonds	NCEP Step 3 Diet No Almonds	*p*-Value
	**% Change from Baseline to 4 Weeks**		
Non-high-density lipoprotein	−4.9	3.5	0.024
Very-low-density lipoprotein	−3.2	12.7	0.01
Intermediate density lipoprotein	−12.4	16.5	0.004
Total cholesterol	−3.4	1.0	0.102
Low-density lipoprotein	−5.3	1.8	0.068
High-density lipoprotein	0.6	−4.2	0.347
Triglycerides	−0.7	11.2	0.068
Lipoprotein (a)	−2.6	−0.1	0.109
Pattern A	−4.0	0.0	N/A
Pattern AB	0.0	0.0	N/A
Pattern B	4.0	0.0	0.003

**Table 6 nutrients-13-01968-t006:** Difference between almond-supplemented (estimated 60 g/day) and almond-free healthy diabetes diets on cardiovascular and glycemic risk biomarkers over 24 weeks [56].

Outcome	Mean Difference	95% Confidence Interval	*p*-Value
Total cholesterol (mg/dL)	−13.9	−22.3, −5.7	0.05
Low-density lipoprotein (mg/dL)	−8.2	−14.7, −1.7	0.011
High-density lipoprotein (mg/dL)	−0.9	−2.7, 0.8	0.301
Triglycerides (mg/dL)	−21.0	NA	0.002
Very-low-density lipoprotein (mg/dL)	−5.6	NA	0.005
Fasting blood glucose (mg/dL	−3.6	−11.9, 4.9	0.411
HemoglobinA1c (%)	−0.3	−0.6, −0.01	0.041
hs-C-reactive protein (mg/L)	−0.3	NA	0.011
Pulse wave velocity (meter/second)	−0.3	NA	0.061
Diastolic BP (mm Hg)	−0.8	−3.1, 1.5	0.402
Systolic BP (mm Hg)	−1.2	−5.1, 2.7	0.530

**Table 7 nutrients-13-01968-t007:** Effect of 500 kcal deficit/day almond vs. control LCDs on cardiometabolic health biomarkers [63].

Outcome	Almond-Enriched Diet	Nut-Free Control Diet	*p*-Value
**Total cholesterol (mg/dL)**			
6 months	−8.7 ± 2.8	−0.1 ± 2.8	0.03
18 months	3.7 + 3.5	5.8 + 3.1	0.64
**Low-Density Lipoprotein Cholesterol (mg/dL)**			
6 months	−5.4 ± 2.9	−0.2 ± 2.9	0.21
18 months	−3.1 ± 2.7	−0.1 ± 2.5	0.41
**High-Density Lipoprotein Cholesterol (mg/dL)**			
6 months	0.4 ± 1.1	−0.6 ± 1.1	0.52
18 months	4.6 ± 1.7	2.3 ± 1.6	0.32
**Total: High-Density Lipoprotein Cholesterol**			
6 months	−0.2 ± 0.1	0.04 ± 0.1	0.02
18 months	−0.2 ± 0.1	−0.1 ± 0.1	0.52
**Triglycerides (mg/dL)**			
6 months	−12.1 ± 4.6	1.0 ± 4.6	0.048
18 months	−4.1 ± 6.4	−10.3 ± 5.6	0.47
**Very-Low-Density Lipoprotein Cholesterol (mg/dL)**			
6 months	−2.4 ± 1.5	1.4 ± 1.5	0.07
18 months	2.3 ± 1.6	3.5 ± 1.4	0.58
**Systolic Blood Pressure (mmHg)**			
6 months	−3.9 ± 1.6	−5.7 ± 1.7	0.44
18 months	−3.2 ± 2.1	−3.6 ± 2.0	0.89
**Diastolic Blood Pressure (mmHg)**			
6 months	−0.8 ± 0.9	−1.6 ± 1.0	0.56
18 months	0.7 ± 1.1	−1.3 ± 1.0	0.19

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
