# Peer review of "A Comprehensive Review of Almond Clinical Trials on Weight Measures, Metabolic Health Biomarkers and Outcomes, and the Gut Microbiota"

_nutrients, 2021, doi:10.3390/nu13061968_

Round 1

Reviewer 1 Report

nutrients-1213705

The ms by Dr. Dreher is a comprehensive review of the literature including meta-analyses and other reviews on the potential health benefits of almond consumption. The ms is very well-written and organized. I have some comments/concerns that I hope serve the author well for further consideration.

Major Concerns

A) Unfortunately, the ms confirms to a convention that has been wrong for too long and refers to body mass as “weight”. Kg are a unit of mass NOT weight and weight is a measure of force that is dictated by gravity. If an individual flies into space there weight is negligible but their mass does not change. The author should correct this mistake and refer to body mass where the use of “kg” is used as units. The author does correctly refer to fat mass and body mass index, but weight and mass are not interchangeable and distinct variables.

B) While the summarization of individual studies is commendable and the amount of work performed to accomplish this task is applauded, a collective summary of uniform data from among the studies where possible would be very useful to get the authors expert interpretation. This is done is some places but the conclusion simply reiterates the major findings without much interpretation.

C) the conclusions in some cases seem a bit too biased. For example, overwhelmingly the data demonstrate that the changes in body mass are modest, and while in some cases statistically significant, overall they are not likely to be biologically significant. For example, larger reductions in body mass than those mentioned here are needed to result in significant reductions in SBP. It is not known if the observed reductions in body masses reported translate to other benefits. This would be much more powerful if the author could calculate some potential relevance of these modest changes. Nonetheless, while the author does refer to these changes as “slight” in the conclusions, a more meaningful interpretation of this and some of the other outcomes would make this review more attractive. An alternative explanation for example could be that given that almonds and other nuts are rich in lipid and potentially increase the risk of unintended body mass gain; however the wealth of data demonstrates that increased adiposity is not a risk with chronic consumption. This is one example. A similar argument can be made for the interpretation of DBP. Not considered the primary risk factor for CVD yet no mention that almonds have essentially no effect on SBP, MAP, and/or pulse pressure to suggest that modification of arterial pressure is a benefit of the more important factors from arterial pressure. Nonetheless, the substantial reduction in the risk of CHD (as presented) is likely due to the beneficial modifications in lipid profiles that are more consistently observed (although still very modest in most cases) because SBP or MAP are not sufficiently improved to provide a biological benefit. Interpretation of the data along these lines would be more impactful. Another example of bias is in Figure 15. The author argues that there is a trend for a reduction in BM even though the numbers are essentially the same and not changed. The figure is not useful here.

Minor

Figures would benefit from more consistency in presentation. For example, many figures just mention that they represent “change from baseline” but the units are not provided. There should be units for all dependent variables. Also, most figures do not denote the significance on the figure even though it is mentioned in the legend (and shown on other figures). More consistent presentation of data in the figures would make it easier to follow.

I would suggest that the author avoid using abbreviations in tables and spell-out each in either a footnote, in the heading, or in the column.

Author Response

Dear Editors and Reviewers,

Thank you for reviewing the almond manuscript and providing critical feedback on this paper. I appreciate your insights and recommendations and believe they have strengthened the quantity of the paper.  In addition to the reviewer’s edits, I made additional editing to clarify content, where possible.  I have uploaded the revised manuscript with tracking changes. 

I believe all your concerns have been addressed but please let me know if there is anything more that I can do to improve the quality of this paper. 

Sincerely, Mark L. Dreher

Response to Reviewer # 1

Major concerns

Thank you for your excellent insights and I hope that I addressed your important concerns in my major revisions. I also made complementary changes throughout the paper including the abstract to reduce bias and promote better flow.

  (A)The manuscript incorrectly refers to weight as equal to body mass.

Body weight was changed to body mass throughout the manuscript text, figures and tables. 

 (B)Collective summaries in the Results Section often reiterated the major findings without much interpretation.

The reiteration of major findings in the overview of the results at the beginning of each sub-section were replaced and augmented by brief interpretive summaries and/or short overviews of the studies.

  1. Conclusions and other sections of the paper in some cases seemed a bit too bias.

(1) I believe that I addressed your major concerns by clarifying the level of importance of the findings and removing the reiteration of the results and potential bias language from the discussion and conclusion section. I tried to include more interpretation and new insights or addressed information that was not previously included in the results section which I thought might be useful for future research.

(2) I deleted Figure 15 and ten other figures especially from lower quality studies and those that were unnecessary or somewhat redundant. 

Minor Concerns

 (A)Need more consistency of figure presentations to make them easier to follow.

Adjustments were made to all the figures and the total number of figures were reduced by 11, which should make the figures easier to follow.

 (B)Avoid abbreviations in tables

The abbreviations were removed from the tables and one very complex table was deleted with its key points were summarized in the results text.  

Reviewer 2 Report

In nutrients-1213705 titled “A Comprehensive Review of Almond Clinical Trials on Weight Measures, Metabolic Health Biomarkers and Outcomes, and the Gut Microbiota” by Mark L Dreher, the author has reported that some randomized clinical trials (RCTs) have beneficial effects on endothelial health, blood flow, hemoglobin A1c to better manage cardiovascular and type 2 diabetes. Recent RCTs suggest possible emerging health benefits for almonds such as enhanced cognitive performance, improved heart rate variability under mental stress, and reduced rate of facial skin aging from exposure to ultraviolet B radiation. Eight RCTs show that almonds could support colonic microbiota health in children and adults by promoting microflora richness and diversity, increasing the ratio of symbiotic to pathogenic microflora, and concentrations of health-promoting colonic bioactive. I have a few concerns regarding the present manuscript.

-The length and the topics that the author has included are impressive, several figures were reported. My first appreciation of the present manuscript is the author needs to add more information in the material and method section, how the manuscripts were selected, which databases were searched, the timeline, and also, how the articles were included or excluded. Then, the quality of the different articles that finally were included (RevMan, Cochrane Library tools).

-It is possible to mix some figures to reduce the original number

-The total data that the author has described is really good, all the topics that you could think of are in the manuscript, it is awesome. When the quality of the selected articles will be measured, some data could be rewritten. 

Author Response

Dear Editors and Reviewers,

Thank you for reviewing the almond manuscript and providing critical feedback on this paper. I appreciate your insights and recommendations and believe they have strengthened the quantity of the paper.  In addition to the reviewer’s edits, I made additional editing to clarify content, where possible.  I have uploaded the revised manuscript with tracking changes. 

I believe all your concerns have been addressed but please let me know if there is anything more that I can do to improve the quality of this paper. 

Sincerely, Mark L. Dreher

Response to Reviewer # 2

Thank you for your excellent insights and I hope that I addressed your important concerns in my revisions.

  (1)Add more details to the Material and Methods Section

I revised the Materials and Methods section to address your concerns to the best of my ability to address inclusion and exclusion criteria.

  (2)Cut out some figures to reduce the original number.

I deleted 11 figures from the manuscript which were primarily from lower quality studies or somewhat redundant.

  (3)Reduced focus on lower quality will improve the quality of the manuscript.

I made major revisions throughout the manuscript in the results and discussion and conclusion section and followed your advice to focus on higher quality RCTs and reduce the number of figures.

Round 2

Reviewer 1 Report

the ms is improved and the changes much appreciated. the flow is better and the edits helped I believe. 

I only have a few comments that the author may want to consider: 
1) I would agree w/ the the strengths and limitations, and there are probably a couple more for each especially the strengths (I think the author may be too modest here, but certainly there are more strengths than limitations here) that could be added. Certainly one limitation that may be important to mention but may not be properly written here (should be re-written in the author's own words) is the lack of consistency among the data which is probably the consequence of the heterogeneity of the study populations and as such quantification of the benefits and sometimes the lack of detected changes should be conditionalized based on the study population. that is there should be some caution in generalizing outcomes for any functional food (almonds here) and the proper interpretation will be a function of the specific study. the same is true when trying to interpret the lack of a change in RCTs where it is very possible there were changes that were not captured by study design or sampling protocol. What if almonds are having much more profound benefits that are not being captured bc of excessive heterogeneity of the study population or being masked by other variables. this leads to my next pt that the author may want to include a brief section on potential future directions that could include ideas for potential improvements in study protocols or design to maybe help improve being able to detect more profound changes.  
2)  I would still like a little better interpretation. the author certainly did a nice job of summarizing the changes, but what do they mean biologically? for example, in Section 4.3 (lines 1602-1606), the author does a nice job of providing a summary of the major effects compiled from the studies w/ mention of essentially no change in arterial pressure. How would the author interpret the biological "significance" of these data collectively. that is, almonds seem to have a pretty consistent benefit on improving lipid profiles but that doesn't translate to improvements in arterial pressure but the rather profound improvements in AI could certainly be a reflection of this (in theory it has to bc of how AI is calculated). there are other examples but hopefully this helps to guide the author. another example that could use the author's interpretation is lines 751-753 where he concludes that almond interventions for bw 2 hrs and 24 wks provide small to moderate benefits in metabolic outcomes. It would be very interesting if the author could provide some ideas of when w/n that time frame there are greater or lesser impacts or if not consistent but give an idea as to why.
3) figures 1-6: y-axis should be something like "changes in body components or body composition" (again weight is not appropriate) but more so some figures have WC, which is a metric of length, not mass/wt but is a body component. the author may want to consider tailoring each legend title to more accurately depict what is being presented in each figure as some are unique and a uniform axis title may not accurately represent each figure

minor edits:
a)  line 120: should delete the "(weight)" bc this implies that mass and weight are synonymous when they are not here
b) Section 4.2, line 1567, 1569: replace "weight" w/ "mass" 

Author Response

Dear Editors and Reviewer 1

Thank you again for reviewing the almond manuscript and I appreciate your insights, comments and recommendations and believe they have further strengthened the quantity of the paper. I also did some additional editing and have uploaded the revised manuscript with tracking changes. 

I believe your concerns have been addressed but please let me know if there is anything more that I can do to improve the quality of this paper as I had cataract surgery on my right eye yesterday, so my vision is somewhat impaired.

Sincerely, Mark L. Dreher

Major concerns

  (1)The strengths and limitations can be improved and the addition of a section of potential future directions.

I updated the section on strengths and limitations and added a brief section on future RCTs. (see red tracking)

  (2)The interpretation of the studies needs to be more improved,

I appreciated your guidance and tried to provide better insights regarding the effects of almonds on metabolic health biomarkers such as the effects of almonds on blood lipids compared to other biomarkers and the effect of duration on biomarker outcomes (see red tracking)

  (3)Figures 1-6 need changes to the y-axis headings

I made changes to the y-axis for figure 1-6 but also updated most of the other figures where they most needed improvements. I also corrected the lingering weight and WC issues. The figures did not show the red tracking because I had to import them from another word file, however, the figure summary changes show the red tracking.

Minor Concerns

I changed the 2 lingering weight to body mass issues that you identified.

Reviewer 2 Report

Thank you to the authors for taking into account my previous comments regarding the manuscript, I felt that the manuscript improved comparing with the other version, however, some typos are detected, and also the use of abbreviations needs more attention

Author Response

Dear Editors and Reviewer 2

Thank you again for reviewing the almond manuscript and I appreciate your insights, comments and recommendations and believe they have further strengthened the quantity of the paper. I also did some additional editing and have uploaded the revised manuscript with tracking changes. 

I believe your concerns have been addressed but please let me know if there is anything more that I can do to improve the quality of this paper as I had cataract surgery on my right eye yesterday, so my vision is somewhat impaired.

Sincerely, Mark L. Dreher

  (1)Typos

I have reviewed the manuscript for typos, and I think they are corrected now but let me know if you still see some.

  (2)Use of abbreviations

My apologies, I tend to use abbreviations especially for blood lipids and other metabolic biomarkers, so they looked normal to me, and I missed them.  I updated all the tables to remove them and adjusted some of the figures too but there are still some abbreviations in figures because they help the figures align better in the small area allowed in the Nutrient pages.